# Sequential Stochastic Combinatorial Optimization Using Hierarchical Reinforcement Learning

**Xinsong Feng**[1]**, Zihan Yu**[2]**, Yanhai Xiong**[3]**, Haipeng Chen**[3]
[1]UCLA, [2]The University of Hong Kong, [3]William & Mary
xsfeng@ucla.edu, u3634664@connect.hku.hk, {yxiong05, hchen23}@wm.edu

## Abstract

Reinforcement learning (RL) has emerged as a promising tool for combinatorial optimization (CO) problems due to its ability to learn fast, effective, and generalizable solutions. Nonetheless, existing works mostly focus on one-shot deterministic CO, while sequential stochastic CO (SSCO) has rarely been studied despite its broad applications such as adaptive influence maximization (IM) and infectious disease intervention. In this paper, we study the SSCO problem where we first decide the budget (e.g., number of seed nodes in adaptive IM) allocation for all time steps, and then select a set of nodes for each time step. The few existing studies on SSCO simplify the problems by assuming a uniformly distributed budget allocation over the time horizon, yielding suboptimal solutions. We propose a generic hierarchical RL (HRL) framework called wake-sleep option (WS-option), a two-layer option-based framework that simultaneously decides adaptive budget allocation on the higher layer and node selection on the lower layer. WS-option starts with a coherent formulation of the two-layer Markov decision processes (MDPs), capturing the interdependencies between the two layers of decisions. Building on this, WS-option employs several innovative designs to balance the model's training stability and computational efficiency, preventing the vicious cyclic interference issue between the two layers. Empirical results show that WS-option exhibits significantly improved effectiveness and generalizability compared to traditional methods. Moreover, the learned model can be generalized to larger graphs, which significantly reduces the overhead of computational resources.

## 1 Introduction

Combinatorial optimization (CO) problems cover a broad spectrum of application domains, such as social networks [45, 2], public health [17, 29], transportation [7], telecommunications [8], and scheduling [40, 55]. Hence, much attention has been drawn from both the theory and application communities toward solving CO problems. Recently, there have been studies using reinforcement learning (RL) to learn generalizable heuristic policies for CO problems on graphs [24, 39, 14, 5, 26, 6]. The RL policy is trained on a set of previously seen training graphs and generalizes to unseen test graphs of similar structure. This policy is usually represented using graph embedding methods such as structure2vec (S2V) [12] and graph convolutional networks (GCNs) [25]. However, most existing studies focus on one-shot deterministic CO, while research on another broad class of CO problems, which contain *stochasticity* and must be solved *sequentially* over multiple time steps, remains limited.

**Problem Statement** In this paper, we study the sequential stochastic CO (SSCO) problem where we first decide the budget allocation for all the time steps, and then select a combination of nodes for each time step. Given a graph $G = (V, E)$, where $V$ and $E$ represent the set of vertices and edges on the graph, respectively, the number of time steps $T$, and the total budget $K$ for selecting nodes, the task is to decide 1) an allocation $K_1, K_2, \ldots, K_T$ of budget over the time horizon, together with 2) a

sequence of node sets $S_1, S_2, \ldots, S_T \subseteq V$, such that

$$\underset{\substack{K_1,\ldots,K_T \\ S_1,\ldots,S_T}}{\text{maximize}} \qquad \sum_{t=1}^{T} r_t(S_t) \qquad (1a)$$

$$\text{subject to} \qquad \sum_{t=1}^{T} K_t \leq K, \qquad (1b)$$

$$|S_t| \leq K_t, \quad \forall t = 1, \ldots, T, \qquad (1c)$$

$$K_t \in \mathbb{N}, \quad \forall t = 1, \ldots, T, \qquad (1d)$$

$$S_t \subseteq V, \quad \forall t = 1, \ldots, T. \qquad (1e)$$

Here, $r_t(S_t)$ denotes the reward obtained from the nodes in set $S_t$ at the time step $t$. SSCO has a wide range of real-world applications such as resource allocation in wireless communication networks [32], route planning [22], and adaptive influence maximization (IM) [42, 48].

As far as we know, there are not many works that address SSCO using RL; two notable examples are adaptive IM in [10] and medical resource allocation for epidemic control in [41]. However, they assume that the budget is evenly allocated across all the time steps (i.e., $K_t = K/T, \forall t$), which may not be optimal in stochastic environments or when additional information or feedback becomes available over time. These scenarios need adaptive sequential decision-making. For example, in the adaptive IM problem, an even allocation policy cannot adapt to sudden changes and uncertainties, leading to resource shortages at critical times or redundancy at less important times. Additionally, it lacks the flexibility to adjust strategy when more important information becomes available.

Despite its criticality, solving SSCO is extremely challenging due to the following factors: 1) *Combinatorial complexity*. The search space grows exponentially as the graph grows, which makes it difficult to find the optimal solution efficiently. 2) *Stochasticity*. Future states and rewards are uncertain, so the ability to adaptively update the solutions based on the real-time state and stochasticity in the future is required. 3) *Sequential decision making*. Decisions are interdependent and require sophisticated planning that considers both current and future consequences. 4) *Scalability*. The methods need to have the ability to be generalized to large-scale problems in the real world.

While traditional RL struggles with these challenges, hierarchical RL (HRL) presents a viable solution by decomposing complex tasks into simpler sub-tasks to improve learning efficiency. Several works have contributed to the development of HRL. The option framework models temporally extended actions, allowing agents to learn at different levels of abstraction [47]. Multi-level approaches use higher-level managers to set tasks for sub-managers [13]. Data-efficient methods leverage offline experiences for training [38]. Combining HRL with intrinsic motivation [27] and hindsight experience replay (HER) [1] further enhances learning.

In this paper, we propose an HRL framework, WS-option, for solving SSCO by jointly considering budget allocation and node selection, with a coherent definition of MDPs for both layers. The lower layer is particularly challenging due to the need for explicit definitions of states and rewards. The MDP formulation directly affects the overall performance of the model. Maintaining model stability is crucial due to the interdependence and mutual influence of the two layers. To improve model stability, especially in the higher layer, we adopt MC methods to learn budget allocation, known as the option policy. To ensure faster convergence and stable operation of the lower layer, we use TD methods to learn node selection, known as the intra-option policy [3]. These methods ensure that both layers can run stably and efficiently, avoiding the issue of potential vicious mutual interference.

Our main contributions are as follows.

- We are the first to formally summarize and define the generic class of SSCO problems.
- We design a novel HRL algorithm, WS-option, to solve the formulated SSCO, with major novelties in terms of the two-layer MDP formulation, and new wake-sleep training procedures to balance the training stability and efficiency and avoid the potential vicious mutual interference. Our algorithm is a generic framework that applies to a wide range of SSCO problems of the same nature. For completeness, we also provide a brief theoretical analysis of the algorithm's convergence.
- We conduct experiments on two distinct SSCO problem instances – adaptive IM and route planning. The results show that compared to traditional methods, our algorithm exhibits superior performance in solving both SSCO problems. In particular, the algorithm is able to generalize to larger unseen graphs, thus significantly reducing computational overheads for larger-scale SSCO problems.

## 2 RELATED WORK

**RL for CO** RL has been widely used to solve CO problems [35]. Nazari et al. [39] propose a generic RL framework for solving the vehicle routing problem (VRP), outperforming traditional heuristic algorithms. Bello et al. [4] introduce neural combinatorial optimization, combining neural networks and RL to solve the TSP problem. Kool et al. [26] develop an attention-based model using the REINFORCE algorithm, achieving superior results on various CO problems like TSP, VRP, and the Orienteering Problem (OP). Khalil et al. [24] combine RL with graph embedding to construct solutions for CO problems step by step. Chen & Tian [11] present a method where neural networks iteratively improve combinatorial solutions through learned local modifications. Deudon et al. [14] introduce a method where policy gradient techniques are used to train neural networks to develop heuristics for solving the TSP. Emami & Ranka [15] propose Sinkhorn policy gradient methods to learn permutation matrices for CO problems. Cappart et al. [9] combine decision diagrams with deep RL to enhance optimization bounds for combinatorial problems. Lu et al. [34] introduce an iterative approach that leverages REINFORCE to enhance solutions for VRPs. Recent methods have further integrated search algorithms, such as active search [20], Monte Carlo tree search [16], and beam search [28], to enhance the solution qualities of RL algorithms during inference time. While these algorithms show promising results, they typically tackle one-shot deterministic CO problems without sequential decision-making, limiting their effectiveness for more complex scenarios.

**Graph embedding** Graph embedding techniques are crucial for RL-based solutions to CO problems, representing graph structures in continuous vector spaces to preserve structural properties, facilitate downstream RL tasks, and generalize to unseen (potentially larger) graphs. Node2Vec [18] uses biased random walks to explore neighborhoods efficiently, which maps nodes to low-dimensional vector spaces while preserving the neighbor structures. Deepwalk [43], which is inspired by language models, uses truncated random walks to obtain local node information and learn potential representations. Significant progress has also been made with graph neural networks (GNN) [44, 58, 53], such as GCN [25], Graph Attention Networks (GAT) [49] and Graph Transformer Networks [57]. Xu et al. [54] evaluate the expressiveness of GNNs, comparing them to the Weisfeiler-Lehman test in capturing graph structures. Li et al. [31] analyze the effectiveness and mechanisms of GCNs in semi-supervised learning. Ying et al. [56] introduce a method to interpret GNNs by identifying the crucial subgraphs and features that influence their predictions. Nonetheless, determining the most beneficial graph embedding technique for RL tasks remains an important research question.

**Hierarchical reinforcement learning** HRL introduces a hierarchical structure to RL, decomposing complex tasks into simpler subtasks to improve learning efficiency. The option framework [47] defines HRL with options as temporally extended actions, which enables agents to learn at different levels of temporal abstraction. Dayan & Hinton [13] introduce a multi-level RL approach based on subgoals, where high-level managers set tasks for sub-managers to achieve efficiently. Nachum et al. [38] propose a data-efficient HRL method using off-line experiences to train higher-level and lower-level layers, and the results demonstrate its ability to learn complex behaviors. Bacon et al. [3] develop a method to automatically learn policy, termination function, and intra-option policy in HRL using intra-option policy gradient and termination gradient, without extra rewards or subgoals. Vezhnevets et al. [50] divide agent behavior into two levels: the Manager, which sets goals in a latent space, and the Worker, which executes these goals with primitive actions. Kulkarni et al. [27] combine hierarchical structures and intrinsic motivation to improve learning in complex tasks. Andrychowicz et al. [1] introduce a technique called HER that enhances RL by learning from failures as if they were successes with different goals, and Levy et al. [30] integrate HER with HRL to enhance learning at multiple levels. Despite their success, these methods often do not address the specific challenges of SSCO problems, especially the need to handle sequential, combinatorial state-action spaces, and dynamic environments. Furthermore, they typically do not integrate graph-based representations, which could significantly enhance their effectiveness when dealing with graph-based CO problems.

## 3 THE WS-OPTION FRAMEWORK

This section will show how to learn budget allocation and node selection using our proposed WS-option framework. We begin with a brief introduction to the option framework, followed by the architecture used in our approach. Subsequently, we will formulate the MDPs for both hierarchical layers and adopt value-based methods as the backbone RL approach. In particular, to balance training

stability and computational efficiency, we employ MC methods to learn the Q-function for the higher layer and TD-learning to learn the Q-function for the lower layer. Additionally, it is important to note that our model is trained on different graphs that follow a certain distribution to enhance its generalization ability to unseen, similarly-structured graphs from the same distribution. The network architecture is shown in Appendix C.1.

### 3.1 PRELIMINARY: THE OPTION FRAMEWORK

The option framework, proposed by [47], is an implementation of temporal abstraction in RL. It provides a straightforward way to describe temporally extended actions, known as options, which are higher-level actions composed of several primitive actions. Formally, an option $o$ is defined by the tuple $(I, \pi^{II}, \beta)$, where

- Initiation set $I$: a set of states where the option can be initialized. We will assume that any state satisfies $s \in I$ in our problems.
- Intra-option policy $\pi^{II}$: a policy that maps states to actions under a given option, controlling the lower-level actions taken under that option. We use $\pi^{II}$ as the lower-level node selection policy.
- Termination function $\beta$: a mapping $\beta : S \to [0, 1]$ that specifies the probability of terminating the option in a given state, where $S$ is the state space.

When an agent operates within the option framework, it follows the process: 1) Option selection – the agent selects an option based on its option policy given a state. 2) Option execution – once the option is determined, the agent follows the intra-option policy $\pi^{II}$ until the option terminates. 3) Option termination – when the termination condition is met, the agent terminates the current option and proceeds to the next option selection.

In our study, we adopt a slightly different approach to defining options. While at the higher layer, the option $o_t^I$ we select represents the budget $K_t$ allocated at the current time step $t$, we simplify the information representation when passing the option to the lower layer. Specifically, we use option $o_t^{II} = 1$ to indicate that the lower layer should continue selecting nodes, and option $o_t^{II} = 0$ to indicate that it should stop selecting nodes and interact with the environment. It is important to note that to handle cases where the budget allocated by the higher layer is 0 (i.e., $o_t^I = o_t^{II} = 0$), we introduce a *null action* $a_{t,\emptyset}$ for the lower layer. The null action $a_{t,\emptyset}$ represents directly interacting with the environment and moving to the next time step. In fact, we achieved very good results by introducing this null action.

### 3.2 HIERARCHICAL MDP FORMULATION

Here, we define the MDPs for the two layers of our hierarchical model (c.f. Figure 1). For ease of understanding, we will take the adaptive IM problem as a running example. More details of both problems are shown in Appendix C.2 and C.3, respectively.

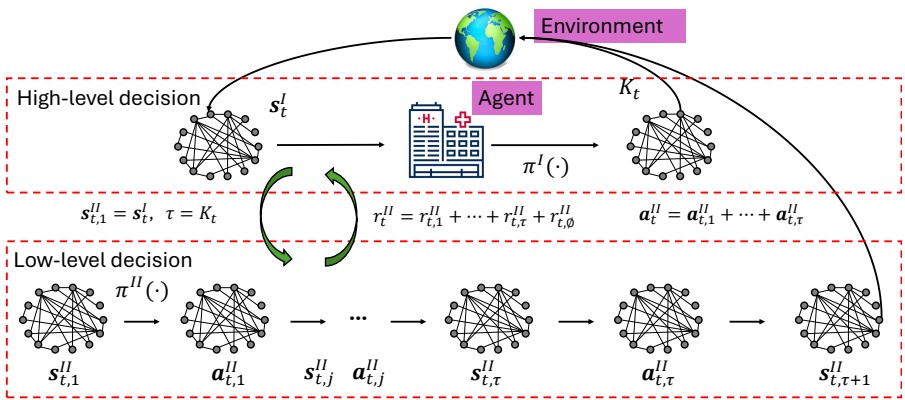

Figure 1: Hierarchical MDPs for SSCO

### 3.2.1 Higher layer MDP

**State**  The state $s^I = (X, g) \in S$ includes two types of information: node features $X$ (e.g., inactive or active in adaptive IM problem) and global information $g$ (e.g., the remaining time steps $T_r$ and the remaining budget $K_r$).

**Option**  The option $o^I$ represents the budget $0 \le K_t \le K_r$ allocated to the current time step $t$.

**State transition**  The state transition in this layer is stochastic, which is a fundamental characteristic of SSCO problems. Given the current state $s_t^I$ and the action $a_t^{II}$ from the lower layer, the state $s_{t+1}^I$ is updated according to the following function:

$$s_{t+1}^I = f(s_t^I, a_t^{II}) + \eta(s_t^I, a_t^{II}), \tag{2}$$

where $f(s_t^I, a_t^{II})$ is a deterministic function that updates the state $s_t^I$ based on the current action $a_t^{II}$, while $\eta(s_t^I, a_t^{II})$ represents the stochastic component, which introduces randomness into the state transition, thereby capturing the stochastic nature of the environment. For example, in the adaptive IM problem, there is no deterministic component. Instead, the entire state transition is governed by the stochastic process $\eta(s_t^I, a_t^{II})$, which can be interpreted as the "activation process" within this problem.

**Reward**  The total reward depends on the problem of interest. In adaptive IM, it is the total number of influenced nodes (e.g., active and removed nodes). Then, the immediate reward $r^I$ is the increase in the total reward.

### 3.2.2 Lower layer MDP

**State**  The state $s^{II}$ is also defined as $s^{II} = \{X, g\}$.

**Option**  The option $o^{II}$ can be either 0 or 1, where "1" represents continuing to select nodes, and "0" represents not selecting any node and interacting with the environment.

**Action**  Two kinds of actions are defined. Given the current time step $t$, the action $a_{t,j}^{II}$ is the node selected for a sub time step $j$ (e.g., $o_{t,j}^{II} = 1$) or just the null action $a_{t,\emptyset}$ (e.g., $o_{t,j}^{II} = 0$). And the $a_t^{II}$ is all the nodes selected in the current time step or the null action $a_{t,\emptyset}$ if no nodes are selected.

**State transition**  The state transition can occur either in a deterministic or stochastic way. If option $o_t^{II}$ is 1 and the action is node $v$, we update the state of node $v$ deterministically (e.g., activating node $v$). If option $o$ is 0, the agent interacts with the environment as described in the higher layer MDP.

**Reward**  The reward in the lower layer significantly affects the convergence of the model. We first define the reward $r_{t,\emptyset}^{II}$ for the null action $a_{t,\emptyset}$, which means directly interacting with the environment at time step $t$. $r_{t,\emptyset}^{II}$ depends on the specific problem, and the intuition is to determine the expected reward when moving to the next day if we do not select any node in the current state. For propagation problems, we can choose $r_{t,\emptyset}^{II} = r_t^I - k_{\text{eff}}$, where $k_{\text{eff}}$ represents the effective number of nodes in action $a_t^{II}$ (e.g., number of inactive nodes in $a_t^{II}$) that can yield actual gain. For problems without a propagation process (e.g., TSP, RP), we can simply set this reward to 0. Next, we define the marginal reward $m_{t,v}$ for node $v$. We denote the selected nodes in the current sub time step $j$ as $A_t = \{a_{t,v_1}, ..., a_{t,v_{j-1}}\}$. The marginal reward is obtained as follows

$$m_{t,v} = R_t(s_t^I, A_t \cup \{a_{t,v}\}) - R_t(s_t^I, A_t), \tag{3}$$

where $R_t(S, A)$ is the immediate reward from taking action $A$ in state $S$. In our experiments, this reward is estimated by averaging the results of multiple (typically 10) simulations. As a result, the reward for selecting node $v$ is given by

$$r_{t,v}^{II} = \frac{m_{t,v}}{\sum_{u \in a_t^{II}} m_{t,u}} (r_t^I - r_{t,\emptyset}^{II}). \tag{4}$$

In this manner, the total reward at each time step for the lower layer will remain consistent with the higher layer, enhancing the model's stability.

### 3.3 WS-OPTION

Due to the high interdependencies between the learned Q-values across the two policy layers, along with advantages such as reduced sample complexity, we employ a value-based RL for both layers. In our framework, the policies of both layers are $\epsilon$-greedy.

---

**Algorithm 1** WS-option: Overall training
___

1: Initialize experience buffer $\mathcal{M}_1$ and $\mathcal{M}_2$ for the higher and lower layer, respectively.
2: Initialize Q-networks for both layers.
3: **for** epoch $i = 1, .., N$ **do**
4:     Sample a Graph $G$ from the distribution $\mathbb{D}$.
5:     **for** episode $j = 1$ **to** $M$ **do**
6:         **if** $i \leq N/2$ **then**
7:             Set the flag `use_fixed_higher_layer` to `true`.
8:         **else**
9:             Set flag `use_fixed_higher_layer` to `false`.
10:         **end if**
11:         Run an episode with the flag `use_fixed_higher_layer` (see Algorithm 2).
12:         Store transitions (see Algorithm 3).
13:     **end for**
14:     Sample a sequential minibatch of the most recent transitions from the replay buffer $\mathcal{M}_1$.
15:     Train the higher layer using MC methods with sampled transitions.
16:     Sample a random minibatch of transitions from the replay buffer $\mathcal{M}_2$.
17:     Train the lower layer using TD-learning with sampled transitions.
18: **end for**

---

**Algorithm 2** WS-option: Run episode
___

1: Initialize the state $s_1^I$ and temporary storage $\mathcal{T}$ for episode transitions.
2: **for** step $t = 1$ **to** $T$ **do**
3:     **if** `use_fixed_higher_layer` is `True` **then**
4:         $o_t^I \leftarrow$ average budget allocation.
5:     **else**
6:         $o_t^I \leftarrow \begin{cases} \text{randomly selected in the remaining budget, w.p. } \epsilon_1, \\ \arg\max_o q^I(s_t^I, o), \text{otherwise.} \end{cases}$
7:     **end if**
8:     Initialize the set of selected nodes $a_t^{II} = \{\}$.
9:     **for** $k = 1$ **to** $o_t^I$ **do**
10:         $s_{t,k}^{II}, o_{t,k}^{II} \leftarrow$ state transition$(s_t^I, o_t^I, a_t^{II})$
11:         $a_{t,k}^{II} \leftarrow \begin{cases} \text{randomly select a node from } V \setminus a_t^{II}, \text{w.p., } \epsilon_2 \\ \arg\max_a q^{II}(s_{t,k}^{II}, o_{t,k}^{II}, a) \text{ subject to } a \notin a_t^{II}, \text{otherwise.} \end{cases}$
12:         $a_t^{II} \leftarrow a_t^{II} \cup \{a_{t,k}^{II}\}$
13:     **end for**
14:     Execute action $a_t^{II}$ and obtain reward $r_t^I$ and next state $s_{t+1}^I$.
15:     Store the transition $(s_t^I, o_t^I, a_t^{II}, r_t^I, s_{t+1}^I)$ in the temporary storage $\mathcal{T}$.
16:     Update the current state: $s_t^I \leftarrow s_{t+1}^I$.
17: **end for**

---

Moreover, as shown from the formulation of the MDPs for both layers, we can see that the Q-functions of the two layers are highly interdependent. On one hand, the higher-layer option selection will determine feasible action spaces for the lower-layer node selection; on the other hand, the specific lower-layer node selection will affect how good the higher-layer's selected option is. Hence, we observe that, if we simply adopt the commonly used (double) Q-learning framework [51, 36, 37, 19], or more essentially off-policy TD learning [46] for both layers, then the Q-values for both layers, especially the first layer will tend to diverge. This is because of the estimation bias derived from the deep models (see Appendix C.1) and bootstrapping of TD learning. We can see the result in Figure 2(b). To stabilize the training of both layers, while maintaining computational efficiency, we propose

the following two designs for our training algorithm, i.e., wake-sleep training and layer-wise learning method selection. The pseudocode of the training algorithm is shown in Algorithm 1. During each epoch, a graph is obtained from a certain distribution $\mathbb{D}$ (line 4). We divide all epochs into two stages (lines 6-10), which correspond to the wake-sleep training procedure described in this section. After we run algorithm 2 and 3, we can use the transitions to train both layers (lines 14-16). Note that during the training of the higher layer, transistions are sampled sequentially (line 14). Although the higher-layer policy, based on MC method, is inherently on-policy and does not require stored transitions, we maintain this to ensure consistency across both layers.

Algorithm 2 shows the process of running episodes. We first get the option (lines 3-6), then choose node one by one (lines 8-13). Subsequently, we run Algorithm 3 (see Appendix B.1) to store transitions for training. Algorithm 3 shows how to store transitions, especially for the lower layer. Since the second layer does not interact with the environment after selecting a single node and suffers from sparse rewards, we first obtain the marginal rewards for each transition state through simulation (lines 5-10). To keep the total reward of the lower layer consistent with the higher layer, we scale all the marginal rewards (lines 12-15), which does not affect the relative $Q$ value of the actions.

### 3.3.1 WAKE-SLEEP TRAINING

The primary challenge in HRL lies in ensuring stable training and model convergence. Notably, training both layers together can lead to instability, as the policies of each layer are interdependent and can interfere with one another. To address this issue, we devise a wake-sleep approach to enhance training stability. As we will show in Section 3.3.3, this proposed wake-sleep training procedure is well-aligned with our theoretical analysis. In Figure 2(a), we present the Q value learned by traditional HRL, where both layers are trained simultaneously from the beginning, and our proposed WS-option. Then, we will provide the details of this wake-sleep training procedure.

In the *sleep* stage of this training paradigm, the Q-function of the higher layer is initially frozen and an average budget allocation strategy is used instead. While more advanced strategies could be considered, we choose the simplest one to allow the lower layer to approach convergence under the current high-layer policy. The lower layer's Q-function is sufficiently trained along with the average budget allocation strategy in this stage. Meanwhile, the high layer's Q-function will be pre-trained offline using the trajectories of the average strategy. In the subsequent *wake* stage, both layers are trained online simultaneously, allowing for the fine-tuning and optimization of their interdependent policies, which ultimately lead to convergence.

### 3.3.2 LAYER-WISE LEARNING METHOD SELECTION

MC methods provide unbiased estimates by calculating the expected value via a complete trajectory, offering reliable and accurate results when the sample size is sufficient. In contrast, TD methods may reduce the accuracy of the Q estimation if errors occur during the intermediate process, known as error propagation. Additionally, in practice, TD methods often result in the overestimation of Q values. Hence, using MC methods to provide more reliable Q value estimates for the higher layer can enhance the model stability. Our experimental results show that this trick, combined with wake-sleep training, is one of the key components that makes the HRL training successful. Figure 2(b) shows the Q values learned using MC and TD methods. The Q values from TD methods are monotonic, while those from MC methods are concave.

On the downside, MC methods are usually more data-hungry because they have to wait until the end of a trajectory to get a training sample. Given that the lower-layer Q-learning (which is essentially a subtask that uses RL to solve one-shot CO problems) has been demonstrated to be reliable [24, 39, 14, 5, 26, 6], we choose the off policy TD-learning (Q-learning) framework for the lower-layer Q-function update. TD-learning has a faster convergence speed compared to MC methods. When training in the sleep stage, the rapid feedback of the TD method can help the lower layer converge faster, so that the whole model will be better initialized with more optimized parameters. This in turn will accelerate the co-training in the wake stage.

### 3.3.3 CONVERGENCE ANALYSIS

For completeness of our study and to offer intuition for our framework, we present a concise theoretical analysis of the algorithm's convergence. We first discuss the convergence of the intra-

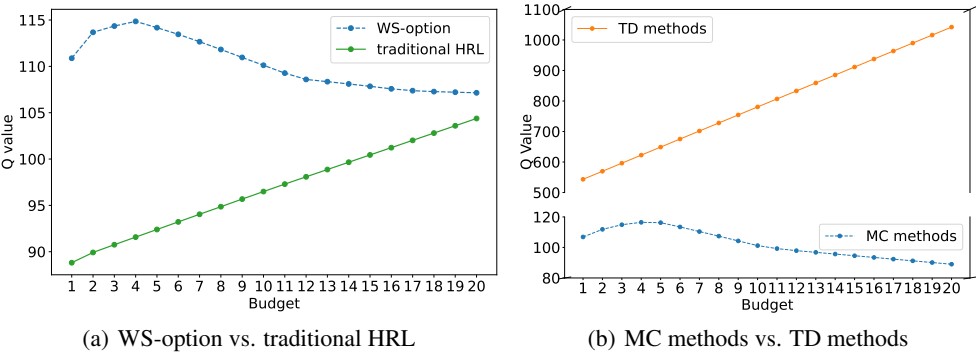

(a) WS-option vs. traditional HRL    (b) MC methods vs. TD methods

Figure 2: Q-values learned for the AIM problem ($T = 10, K = 20$).

option policy and give Theorem 1. Then, we show the convergence of the option policy and give the Theorem 2. The proofs are provided in Appendix A.5.1 and Appendix A.5.2, respectively.

**Theorem 1.** (Intra-option policy convergence). *In our WS-option framework, given any Markov transition $(s_\tau, o_\tau, a_\tau, r_\tau, s_{\tau+1}, o_{\tau+1})$, the Q-value function $q^{II}(s_\tau, o_\tau, a_\tau)$ converges to the optimal Q-value function $q_*^{II}(s_\tau, o_\tau, a_\tau)$ with probability 1, assuming that the higher-layer policy is fixed.*

**Theorem 2.** (Option policy convergence). *In our WS-option framework, given any state-option pair $(s_t, o_t)$, the Q-value function $q^I(s_t, o_t)$ converges to the optimal Q-value function $q_*^I(s_t, o_t)$ with probability 1, assuming that for any given higher-layer policy, the lower-layer policy always provides the corresponding conditionally optimal response.*

As a result, the convergence of the hierarchical framework is guaranteed by the convergence of both layers. Although the interaction between layers will have some effects, we use the wake-sleep procedure instead. As illustrated in Figure 3, after the sleep stage, we can get an optimal policy $\pi^{II}$ conditioned on the higher layer's policy $\pi^I$. Notably, even when the higher layer policies differ, the nature of the node selection task leads to lower layer optimal policy $\pi^{II}$ being similar to any conditional optimal policy $\pi'^{II}$. This is analogous to the Stackelberg game setting, where the follower can always give best response. In this case, the convergence of the total framework can be achieved. Notably, our analysis is just limited to the tabular case.

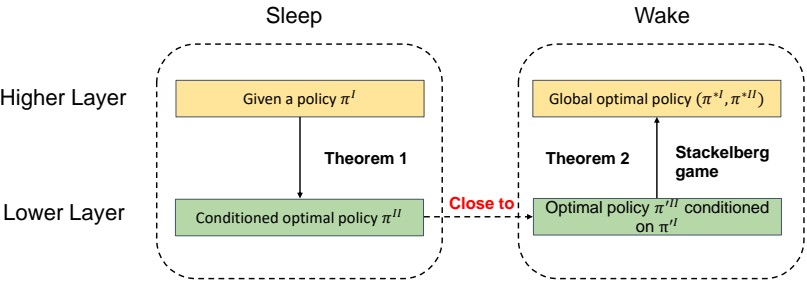

Figure 3: Wake-sleep training procedure

# 4    EXPERIMENTAL RESULTS

## 4.1    SSCO PROBLEM INSTANCES

In this article, we examine two broad classes of problems: the propagation problem (high stochasticity) and the route planning problem (low stochasticity). For the propagation problem, we use an adaptive version of the classic IM problem. For the route planning (RP) problem, we consider a model tailored for travel route planning. Notably, instead of showing the standard deviation of test results, we perform two-sample t-tests to show the effectiveness of our framework in SSCO scenarios.

## 4.2 Comparison with baselines

**Baselines** To evaluate the effectiveness of our approach, we compare it against appropriate baselines for each problem, measuring the cumulative reward obtained by various methods. Each baseline is named in the form of A-B, where A is the higher-layer algorithm, and B is the lower-layer algorithm. For the AIM problem, we use the baselines mentioned in Tong et al. [48]: average policy, static policy, and normal policy at the high level. The static policy divides the time period $T$ into $d$ cycles, allocating the same resources only on the first day of each cycle. The normal policy allocates all resources at the beginning. At the lower level, we use a degree-based greedy strategy and a well-designed score-based strategy. The score for each node $v$ is defined as $s_v = \sum_{u \in \mathcal{N}_{\text{in}}(v)} p(u, v)$, where $\mathcal{N}_{\text{in}}(v)$ denotes the set of inactive nodes among the neighbors of $v$, and $p(u, v)$ is the probability that $u$ can successfully activate $v$. This method takes into account the characteristics of the problem and uses expected values to score each node, showing strong performance in our experiments. For the RP problem, we use genetic algorithms (GA) and greedy algorithms. We train and test on graphs of the same size here. The results are shown in Tables 1-2, where $T$ and $K$ are chosen according to Tong et al. [48].

Table 1: Experimental results for AIM, $n = 200$. All cases have p-values $\leq 0.05$.

| Method | $T, K = 10, 10$ | $T, K = 10, 20$ | $T, K = 10, 30$ | $T, K = 20, 10$ |
|---|---|---|---|---|
| **WS-option** | **76.00** | **118.56** | **129.06** | **80.95** |
| average-degree | 67.92 | 104.54 | 122.50 | 72.18 |
| average-score | 74.36 | 116.10 | 128.29 | 80.31 |
| normal-degree | 69.28 | 101.50.34 | 109.39 | 63.47 |
| normal-score | 75.05 | 111.89 | 118.78 | 70.68 |
| static-degree | 70.02 | 105.25 | 122.37 | 70.57 |
| static-score | 74.81 | 118.13 | 128.01 | 71.68 |

Table 2: Experimental results for RP, $n = 100$. All cases have p-values $\leq 0.05$.

| Method | $T, K = 10, 10$ | $T, K = 10, 20$ | $T, K = 10, 30$ | $T, K = 20, 10$ |
|---|---|---|---|---|
| **WS-option** | **7.46** | **12.86** | **18.57** | **7.52** |
| greedy | 6.29 | 12.02 | 15.68 | 6.73 |
| GA | 6.79 | 11.65 | 15.70 | 6.93 |

As shown in Tables 1-2, our method performs consistently well across various settings. As expected, our method outperforms the baseline methods in the AIM problem, which emphasizes structural information about the graph as well as feedback processing of stochasticity (whether the node is successfully activated or not). In our experiments, we use simple graph neural networks. In fact, employing more advanced graph neural networks could further improve the performance of our algorithm, but graph embedding is not the focus of our study. For the RP, our algorithm also shows advantages. The graph of RP is essentially a fully connected graph, which does not emphasize the structural information of the graph but focuses more on the distance between nodes. Nonetheless, the primary advantage of our algorithm is its ability to be applied to larger graphs without the need for retraining after a single training, which significantly reduces computational overhead compared to traditional algorithms. Additionally, cumulative reward during training for AIM problem is provided in Figure 5 of Appendix D.1.

## 4.3 Assessing the learned policies

We now assess the learned budget allocation and node selection policies using the AIM problem. By fixing one layer's policy, we compare the cumulative rewards obtained by only the other layer, enabling us to independently assess the policy of each layer.

To evaluate the higher layer, we fix the lower layer using its learned policy and vary the budget allocation policy. Conversely, to evaluate the performance of the lower layer, we fix the higher layer using the average budget allocation policy and vary the node selection policy. As shown in Table

Table 3: Cumulative rewards when varying one layer's policy while the other layer remains fixed. All cases have p-values $\leq 0.05$.

| Setting | lower layer fixed (using the learned policy) | | | |
|---|---|---|---|---|
| | **WS-option** | average | normal | static |
| $T, K = 10, 10$ | **76.79** | 71.45 | 75.27 | 74.85 |
| $T, K = 10, 20$ | **127.51** | 126.35 | 120.26 | 125.46 |

| Setting | higher layer fixed (using the average policy) | | |
|---|---|---|---|
| | **WS-option** | degree | score |
| $T, K = 10, 10$ | **71.45** | 62.55 | 69.15 |
| $T, K = 10, 20$ | **126.35** | 118.75 | 125.02 |

3, our algorithm performs well even when applied to a single layer. Both the node selection and the learned budget allocation strategies yield better results compared to the baseline approach. The learned budget allocation policy is showed in Appendix D.2, Figures 6 to 8.

### 4.4 GENERALIZATION TO LARGER GRAPHS

Most excitingly, our algorithm generalizes well to larger graphs. This means that models trained on small graphs can be effectively applied to larger ones. The reason for this generalization ability lies in the framework we use, which does not include any parameters related to graph size or number of nodes. This allows us to use the same parameters for graphs of different sizes.

Table 4: AIM: Generalization to larger graphs. All cases have p-values $\leq 0.05$.

| | 200 | 400 | 700 | 1000 |
|---|---|---|---|---|
| **WS-option** | **120.88** | **176.13** | **204.87** | **221.16** |
| average-degree | 110.18 | 144.93 | 155.85 | 167.58 |
| average-score | 119.80 | 158.98 | 165.24 | 176.31 |
| normal-degree | 103.22 | 152.46 | 186.01 | 210.57 |
| normal-score | 106.47 | 166.45 | 199.41 | 220.81 |
| static-degree | 109.62 | 155.08 | 178.66 | 198.44 |
| static-score | 118.86 | 171.91 | 188.44 | 204.16 |

We train the model on graphs with 50-100 nodes with $T = 10, K = 20$. Table 4 and Table 7 (see Appendix D.3) present the results of our tests on different graph sizes. As shown, even though the model was trained on small graphs, it still exhibits excellent performance on larger graphs. This result demonstrates that our algorithm has good scalability and generalization ability to effectively deal with large-scale graph structure problems.

## 5 CONCLUSION

We propose an option-based HRL framework for solving complex SSCO problems. The core of our approach is to use the wake-sleep training procedure and layer-wise learning method selection to make the system more stable without interfering with each other, as well as to facilitate our learning process by giving a clear definition of MDPs of two layers. Through extensive experimental evaluations, we demonstrate the strong performance of our algorithm in solving the SSCO problem, its ability to learn effective budget allocation and node selection policies, and its strong generalization to larger graphs. However, a notable limitation of our algorithm is that each new problem needs a new graph embedding technique, which adds complexity to the framework's implementation across different scenarios. We leave the investigation of model-agnostic graph embedding techniques for RL-based solutions to SSCO as future work, potentially borrowing ideas of recent advancements on graph foundation models [33].

## ACKNOWLEDGEMENTS

This research was partially supported by the National Science Foundation under IIS-2348405. The authors acknowledge William & Mary Research Computing for providing computational resources and/or technical support that have contributed to the results reported within this paper.

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

CONTENTS

# A    THEORETICAL INSIGHTS

## A.1    HIERARCHICAL LEARNING FOR BI-LEVEL OPTIMIZATION

Based on the problem formulation in equation 1, we provide a high-level insight into the nature of this problem, which motivates the use of HRL. Specifically, the SSCO problem can be reformulated as a bi-level optimization problem:

$$\max_{\pi^h} \quad J(\pi^I, \pi^{II*}(\pi^I)) \tag{5}$$

$$\text{subject to} \quad \pi^{II*}(\pi^I) \in \arg\max_{\pi^I} J(\pi^I, \pi^{II}) \tag{6}$$

where $J = \sum_{t=1}^{T} r_t(S_t)$, $\pi^I = \{K_t\}$, and $\pi^{II} = \{S_t\}$. Here, $\pi^I$ is constrained by equation 1b, and $\pi^{II}$ satisfies the constraints in equation 1c.

The formulation highlights that the SSCO problem inherently exhibits a bi-level optimization structure. The higher-level policy $\pi^I$ determines the budget allocation strategy (e.g., $\{K_t\}$), while the lower-level policy $\pi^{II}$ selects the corresponding subsets (e.g., $\{S_t\}$) based on the higher-level decisions.

Given this hierarchical dependency, it is natural and intuitive to employ a hierarchical learning algorithm such as HRL to address such problems effectively.

In contrast, traditional single-agent RL methods struggle with this bi-level optimization problem due to the following limitations:

- **Hierarchical Dependency**: Single-agent RL lacks the structure to handle the interdependent objective, as the lower-level policy $\pi^{II}$ directly depends on higher-level decisions $\pi^I$.

- **Exploration Efficiency**: HRL separates exploration across two tractable levels, improving learning effiency compared to single-agent RL's flat exploration approach, which is one of the HRL's greatest advantages [52].

- **Scalability and Interpretability**: Optimizing the joint action space of both levels in single-agent RL is computationally intractable. HRL, by decoupling the problem into manageable subproblems, offers a scalable, interpretable, and modular framework for solving complex hierarchical tasks.

## A.2    INTUITION BEHIND WAKE-SLEEP OPTION FRAMEWORK

The core idea for solving this bi-level optimization problem is to decompose it into two separate single-level optimization problems.

When the lower layer provides a conditionally optimal solution, denoted as $\pi^{II*}(\pi^I)$, the bi-level problem reduces to a single-layer problem, which corresponds to the objective defined in equation 5. In this context, the task is just to find the optimal policy for the higher layer. Conversely, when the higher-layer policy is fixed, the problem becomes solving the sub-optimization objective in equation 6, which is to find the corresponding conditionally optimal lower-layer policy. Building on this decomposition, the framework iteratively refines the higher-layer policy by alternating between the two optimization processes. Specifically, a higher-layer policy is first provided, and the corresponding conditionally optimal lower-layer policy is determined. This iterative procedure continues until the global optimal solution is achieved.

However, in the context of the SSCO problem, the role of lower layer is to perform node selection, while the higher layer only adjusts the termination condition. Due to the nature of this task, we claim that the conditionally optimal lower-layer policy remains similar across different higher-layer policies $\pi^I$. Given this, the wake-sleep procedure is employed. During the sleep stage, the lower layer determines a near-optimal policy. Subsequently, the two layers are trained jointly during the wake stage. Since the lower layer has already converged to a close-to-optimal policy, the joint training further refines both the higher-layer policy and the lower-layer policy, ensuring that the global optimal solution is achieved.

### A.3    SIMPLIFICATION OF OPTIONS IN THE LOWER LAYER

In this section, we justify our simplification of options and provide a deeper understanding of the option framework within our HRL approach. First, we revisit the definition of an option $o$ in 3.1. In the traditional option framework, an option $o$ is defined as a tuple $(I, \pi^{II}, \beta)$, where:

- $I$ is the initiation set, which specify the states where the option can be initialized.
- $\pi^{II}$ is the lower-policy that the agent follows while the option is being executed.
- $\beta$ is the termination condition that determines when the option should terminate.

Options allow the agent to execute temporally extended actions, thereby enabling more efficient exploration and decision-making by abstracting lower-level actions into higher-level policies.

#### A.3.1    WHY SIMPLIFY OPTIONS?

In our specific bi-level optimization problem, the primary role of the option in the lower layer is to determine when the lower-level policy $\pi^{II}$ should terminate its node selection process. Specifically, the higher-level budget allocation $K_t$ implicitly decides the number of nodes the lower layer should select before terminating. Given this context, we simplify options due to the following reasons.

**Redundant Information and Enhanced Interpretability**    The numerical value of the budget $K_t$ does not provide additional useful information beyond determining the number of node selections. Therefore, conveying the exact budget value to the lower layer is unnecessary and does not contribute to the termination decision. By representing options as binary indicators, we clarify whether the lower layer should continue or terminate, enhancing the interpretability of the policy.

**Improved Learning Efficiency**    Reducing the option to a binary indicator allows the lower layer to focus solely on a straightforward termination condition, without needing to interpret specific budget values. This simplification minimizes the complexity of the learning task, leading to faster convergence and more stable learning processes.

Therefore, we simplify the option passed to the lower layer to a binary indicator:

- $o_t^{II} = 1$: Indicates that the lower layer should continue selecting nodes.
- $o_t^{II} = 0$: Indicates that the lower layer should stop selecting nodes and interact with the environment.

#### A.3.2    JUSTIFICATION OF BUDGET IRRELEVANCE

It is important to note that the specific numerical value of the budget $K_t$ does not inherently affect the termination condition beyond determining the number of node selections. The budget serves as a constraint that limits the range of the lower layer's actions but does not need to be explicitly taken as part of the option. By abstracting the budget into a binary termination signal, we retain the essential functionality required for effective decision-making without introducing additional complexity.

Overall, this design choice strikes a balance between simplicity and functionality, leveraging the hierarchical structure of our problem to enhance learning efficiency and policy interpretability.

### A.4    LOWER-LAYER REWARD DESIGN

The reward structure in the lower layer is a critical component of the WS-option framework, as it directly influence both the stability and convergence of the model. Proper design of the reward ensures that the hierarchical optimization aligns seamlessly across layers, mitigating potential instability caused by sparse or inconsistent feedback signals.

#### A.4.1    CHALLENGES IN LOWER-LAYER REWARD DESIGN

The task of the lower layer is to select nodes based on the budget allocated by the higher layer. However, if the lower layer were to select all $K_t$ nodes in a single step, the size of the action

space would grow combinatorially as $\binom{N}{K_t}$, where $N$ is the total number of (available) nodes. This combinatorial explosion makes the optimization process computationally infeasible for large $N$ and $K_t$. Therefore, we adopt a sequential selection approach where the lower layer selects one node at a time and repreat this process for $K_t$ times. This strategy effectively reduces the action space size from $\binom{N}{K_t}$ to $N$ per selection step, significantly enhancing the tractability of the optimization problem.

However, this sequential approach introduces a new challenge: sparse rewards. Specifically, the environment only provides feedback after the final node in the sequence has been selected, resulting in delayed and aggregate reward signals. Such delayed feedback make it difficult for the model to identify the contribution of individual node selections to the overall reward.

### A.4.2 NORMALIZATION OF MARGINAL REWARD

We normalize marginal rewards to ensure that the total reward of the lower layer for each time step $t$ is the same as the reward of the higher layer, which is

$$\sum_{u \in a_t^{II}} r_{t,u}^{II} + r_{t,\emptyset}^{II} = r_t^I = R(s_t^I, o_t^I, a_t^{II}). \tag{7}$$

At each time step, the reward $R(s_t^I, o_t^I, a_t^{II})$ is determined jointly by both the higher-layer option $o_t^I$ and the lower-layer action $a_t^{II}$. Since both layers share the same reward and objective, it is essential that the rewards remain aligned.

When the lower-layer actions are decomposed into sequential sub-actions (e.g., selecting one node at a time), the individual rewards for each sub-action must sum up to the total reward $R(s_t^I, o_t^I, a_t^{II})$. Without normalization, the reward distribution across these sub-actions could deviate, leading to inconsistencies between the lower layer's objective and the higher layer's global reward signal. As a result, by normalizing the marginal reward, we ensure: 1) reward consistency. The total reward for the lower layer matches the original reward defined by the hierarchical framework, maintaining coherence between the two layers; and then 2) objective alignment. Both the higher layer and lower layer optimize the same objective, preserving the shared purpose of achieving the overall framework's goals.

### A.5 CONVERGENCE ANALYSIS

### A.5.1 PROOF OF THEOREM 1

**Theorem.** (Intra-option policy convergence). *In our WS-option framework, given any Markov transition $(s_\tau, o_\tau, a_\tau, r_\tau, s_{\tau+1}, o_{\tau+1})$, the Q-value function $q^{II}(s_\tau, o_\tau, a_\tau)$ converges to the optimal Q-value function $q_*^{II}(s_\tau, o_\tau, a_\tau)$ with probability 1, assuming that the higher-layer policy is fixed.*

*Proof.* The update formula of intra-option Q-value in tabular form is as follows:

$$q_{n+1}^{II}(s_{t,\tau}, o_{t,\tau}, a_{t,\tau}) = \begin{cases} r_{t,\tau} + \gamma \max_a q_n^{III}(s_{t,\tau+1}, o_{t,\tau+1}, a), & \text{If } o_{t,\tau} > 0 \text{ and } a_{t,\tau} \neq a_{t,\phi}, \\ r_{t,\tau} + \gamma q^I(s_{t+1}, o_{t+1}), & \text{otherwise.} \end{cases} \tag{8}$$

According to Theorem 1 from [21], we need to show the update operator is a contraction mapping. For $o_{t,\tau} > 0$ and $a_{t,\tau} \neq a_{t,\phi}$,

$$\begin{aligned} &|q_{n+1}^{II}(s_{t,\tau}, o_{t,\tau}, a_{t,\tau}) - q_*^{II}(s_{t,\tau}, o_{t,\tau}, a_{t,\tau})| \\ =&|r_{t,\tau} + \gamma \max_a q_n^{II}(s_{t,\tau+1}, o_{\tau+1}, a) - r_{t,\tau} - \gamma \max_a q_*^{II}(s_{t,\tau+1}, o_{t,\tau+1}, a)| \\ =&\gamma|\max_a q_n^{II}(s_{t,\tau+1}, o_{t,\tau+1}, a) - \max_a q_*^{II}(s_{t,\tau+1}, o_{t,\tau+1}, a)| \\ \leq&\gamma \max_a |q_n^{II}(s_{t,\tau+1}, o_{t,\tau+1}, a) - q_*^{II}(s_{t,\tau+1}, o_{t,\tau+1}, a)| \end{aligned} \tag{9}$$

For the other case,

$$|q_{n+1}^{II}(s_{t,\tau}, o_{t,\tau}, a_{t,\tau}) - q_*^{II}(s_{t,\tau}, o_{t,\tau}, a_{t,\tau})| = |r_{t,\tau} + \gamma q^I(s_{t+1}, o_{t+1}) - r_{t,\tau} - \gamma q_*^I(s_{t+1}, o_{t+1})| \tag{10}$$

Note that $q^I(s_{t+1}, o_{t+1}) = \max_a q^{II}(s_{t+1}, o_{t+1}, a)$. Therefore, we have

$$
\begin{aligned}
&|q^{II}_{n+1}(s_{t,\tau}, o_{t,\tau}, a_{t,\tau}) - q^{II}_*(s_{t,\tau}, o_{t,\tau}, a_{t,\tau})| \\
=&\gamma|\max_a q^{II}_n(s_{t+1}, o_{t+1}, a) - \max_a q^{II}_*(s_{t+1}, o_{t+1}, a)| \\
\leq&\gamma\max_a|q^{II}_n(s_{t+1}, o_{t+1}, a) - q^{II}_*(s_{t+1}, o_{t+1}, a)|
\end{aligned}
\tag{11}
$$

So we prove the convergence of the intra-option policy. $\square$

### A.5.2 Proof of Theorem 2

**Theorem.** (Option policy convergence). *In our WS-option framework, given any state-option pair $(s_t, o_t)$, the Q-value function $q^I(s_t, o_t)$ converges to the optimal Q-value function $q^I_*(s_t, o_t)$ with probability 1, assuming that for any given higher-layer policy, the lower-layer policy always provides the corresponding conditionally optimal response.*

*Proof.* The Q-value of the higher layer is updated by:

$$
q^I_{t+1}(s_t, o_t) = q^I_t(s_t, o_t) - \alpha_t(s_t, o_t)(q^I_t(s_t, o_t) - g_t),
\tag{12}
$$

where $g_t$ is the return estimated by MC methods. Again, according to the Theorem 1 from [21], to establish the convergence, we need to show that $\mathbb{E}[q^I_t(s_t, a_t) - g_t] = 0$. Under the assumption that for any given higher-layer policy, the lower-layer always provides the best-response policy, the lower-layer policy remains fixed and conditionally optimal at each time step $t$. at each time step $t$, the lower policy is fixed and the conditionally optimal. Consequently, the expected value of the observed return $g_t$ matches the Q-value estimate $q^I_t(s_t, o_t)$. Therefore, the condition $\mathbb{E}[q^I_t(s_t, o_t) - g_t] = 0$ is satisfied. As a result, the Q-value function $q^I$ converges to the optimal option Q-value $q^I_*$ with probability 1. $\square$

### A.5.3 Interaction and Convergence of Hierarchical Layers

The theorems presented above establish the convergence of two separate single-level optimization problems, as discussed in Section A.2. However, ensuring the convergence of the entire framework for bi-level optimization remains challenging due to the inherent interdependence between the two layers.

To address this, we alternate between the two optimization processes and further use our WS option framework, which effectively decouples their dependencies during training. Specifically, the higher-layer policy provides guidance to the lower-layer policy, which then adapts to the given conditions and converges to its conditionally optimal response. Once the lower layer converges, the higher-layer policy is refined based on the updated performance metrics.

This iterative approach leverages the individual convergence of each layer, as proven in Theorems in A.5.1 and A.5.2, to ensure the overall framework converges. By combining the independent convergence of the two layers, the framework achieves global optimization for the bi-level problem.

## B  ALGORITHM DETAILS

### B.1  ALGORITHM 3

---

**Algorithm 3** WS-option: Store transitions

---

 1: **for** each transition $(s_t^I, o_t^I, a_t^{II}, r_t^I, s_{t+1}^I)$ in $\mathcal{T}$ **do**
 2:     Compute the gain $g_t = \sum_{i=t}^T \gamma^{i-1} r_i^I$.
 3:     Store the transition $(s_t^I, o_t^I, g_t)$ in the replay buffer $\mathcal{M}_1$.
 4:     Set $s_\tau^{II} \leftarrow s_t^I$ and $o_\tau^{II} \leftarrow o_t^I$.
 5:     **for** each node $u$ in the $a_t^{II}$ **do**
 6:         $s_{\tau+1}^{II}, o_{\tau+1}^{II} \leftarrow$ state transition$(s_\tau^{II}, o_\tau^{II}, u)$
 7:         Estimate the marginal reward $m_\tau^{II}$ by running 10 simulations.
 8:         Store the intermediate transition $(s_\tau^{II}, o_\tau^{II}, u, m_\tau^{II}, s_{\tau+1}^{II}, o_{\tau+1}^{II})$ for later scaling.
 9:         Update $s_\tau^{II} \leftarrow s_{\tau+1}^{II}$ , $o_\tau^{II} \leftarrow o_{\tau+1}^{II}$.
10:     **end for**
11:     **for** each stored intermediate transition $(s_\tau^{II}, o_\tau^{II}, u, m_\tau^{II}, s_{\tau+1}^{II}, o_{\tau+1}^{II})$ **do**
12:         Scale the reward $r_\tau^{II} = \frac{m_\tau^{II}}{\sum m_k^{II}}(r_t^I - r_{t,\emptyset}^{II})$.
13:         Store the transition $(s_\tau^{II}, o_\tau^{II}, u, r_\tau^{II}, s_{\tau+1}^{II}, o_{\tau+1}^{II})$ in the replay buffer $\mathcal{M}_2$.
14:     **end for**
15: **end for**

---

## C  IMPLEMENTATION DETAILS

### C.1  NETWORK ARCHITECTURE

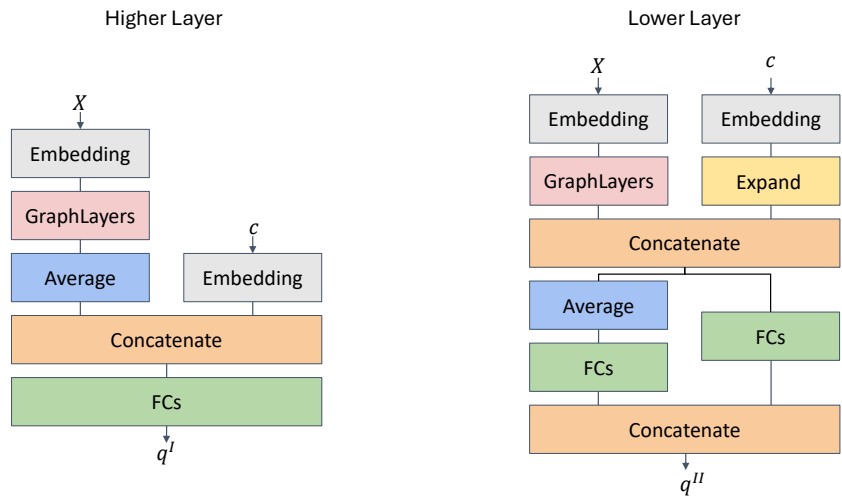

Figure 4: Network architecture

The network architecture consists of two primary components: the higher layer network and the lower layer network, as illustrated in Figure 4. There are two inputs: the node feature $X$ and the context $c$. We use context $c$ to represent non-node feature information, such as global information $g$ and the option $o^I$, which is a one-dimensional vector. The basic idea is to use graph network layers to get node embeddings. Then, we concatenate the node embeddings with the context embedding, which are used as the input of linear layers.

**Higher layer network with action-in**  This network is used for estimating $q^I(s^I, o^I)$. It is worth noting that we adopt an action-in network structure, which differs from the traditional action-out

structure in typical Q-learning frameworks like DQN [36, 37]. In the action-in structure, actions are also taken as inputs. In our study, this means the option (budget) is an input. We choose the action-in structure for two main reasons: first, it can capture the properties of a particular state-action pair more accurately because of the ability to embed the action into potential features. More importantly, since the size of the budget is not used as a parameter of the network (i.e., the number of output nodes), this approach allows for more powerful generalization ability of the model, which is not possible with the action-out structure.

For the network architecture, we use graph network layers to obtain node embeddings $h$. An average readout function is then applied to get the graph embedding $\bar{h}$. After concatenating this with the context embedding, we use linear layers to compute $q^I(s^I, o^I)$.

**Lower layer network with action-out**    The lower layer network adopts a traditional "action-out" structure to estimate $q^{II}(s^{II}, o^{II}, a^{II})$. After obtaining the node embeddings $h$ through graph layers, we concatenate each node embedding $h_i$ with the context embedding $c$ (so we need to expand it first). Subsequently, we average the node embeddings to get the graph embedding, which is used to denote the null action.

**Details of the network**    In our study, we use different architectures for two problems. For the AIM problem, we employ the S2V structure in [24] with $T = 3$ to obtain node embedding and graph embedding (64-dimensional) for the higher layer. Subsequently, we use a linear layer to get the embedding of the option (64-dimensional), which is also called context embedding. We then concatenate the graph embedding with the context embedding and apply three linear layers to get the Q-value. For the lower layer, we also use S2V structure but with $T = 1$ to get both node embedding and graph embedding (64-dimensional). And using two linear layers for node embedding and graph embedding, respectively, to estimate the Q-value of all action ($n$ nodes and a null action). To tackle the RP problem, we use attention layers due to their effectiveness in capturing the features of distances and benefits associated with nodes. For the higher layer, three attention layers with an embedding size of 64 are used to get the node and graph embeddings. Additionally, for this problem, the context is the option, the current node and the starting node. We use a linear layer to get the option embedding and combine it with node embeddings of the current node and starting node to get the context embedding and concatenate them with the graph embedding. The combined embedding is then passed through three linear layers to estimate the Q-value. For the lower layer, a single attention layer is used to get the node embedding (64 dimensional). The context embedding is the same as the higher layer. Notably, the embedding of the current node is employed as the representation for the null action instead of the graph embedding. Finally, three linear layers are applied to get the Q-value.

## C.2    ADAPTIVE INFLUENCE MAXIMIZATION (AIM)

AIM is an extension of IM [23], which aims to progressively select seed nodes in dynamic and uncertain environments to maximize the spread of information within a social network. AIM has diverse applications in fields such as viral marketing, information dissemination, and public health. In this paper, we use the Independent Cascade (IC) model as the information propagation model.

The problem is modeled on a directed graph $G = (V, E)$, where $V$ and $E$ are nodes and edges, respectively. Each node can be either active or inactive. Once a node $u$ becomes active (either by being selected as a seed or being activated by other nodes), it will attempt to activate each of its inactive neighbors $v$ with a probability $p(u, v)$ on the next day and cannot activate nodes afterwards. The probability $p(u, v)$ is constant and is set to $\frac{1}{d^-(v)}$ in our model, where $d^-(v)$ denotes the indegree of node $v$. Initially, all nodes are inactive, and before each day starts, we can select some seed nodes to activate. We assume a total budget $K$ (i.e., the total number of nodes that can be selected) and study how to allocate the seed nodes to maximize the influenced nodes within the time constraint of $T$ days [48].

The MDP formulation of AIM is described in the following. The datasets used in the experiments are randomly generated using the `erdos_renyi_graph()` function from Python's NetworkX library, with a probability of $0.01$.

**MDP formulation**    For the higher level, we denote the state of nodes as $X_t \in \{0, 1\}^{|V| \times 3}$. The state of each node is represented as a one-hot vector, indicating that the node is in one of three possible states: inactive, active (able to activate other nodes), or removed (active but unable to activate

other nodes). What's more, the global information here is the remaining budget. The option is the budget allocated to the current time step. The reward is the increase in the number of influenced nodes. For the state transition, for example, given the current state $s_t^I$ and action $a_t^{II}$ (selected nodes) from the lower layer, we first change the state of selected nodes $a_t^{II}$. For each node $v \in a_t^{II}$,

$$X_t^v = [0, 1, 0]. \tag{13}$$

Now we consider the interaction with environment. Assume the set of current active nodes is $C_{a,t}$ and the set of their inactive neighbors is $C_{ina,t}$. Then, for each node $v \in C_{ina,t}$, the probability of being activated is

$$P(X_t^v = [0, 1, 0]) = 1 - \prod_{u \in \mathcal{N}(v) \cap C_{a,t}} (1 - p(u, v)). \tag{14}$$

where $\mathcal{N}(v)$ is the set of neighbors of node $v$, and $p(u, v)$ is the probability that node $v$ is activated by node $u$. Then, we update the state of newly activated nodes $\mathcal{A}$ as in equation 13.

For the lower level, the state remains consistent with the higher level but includes the option chosen by the higher level. The action is to select a node. The immediate reward is defined in Equation 4. The state transition occurs by simply setting the selected node to an active state.

## C.3 ROUTE PLANNING (RP)

RP includes a broad class of optimization problems that seeks to find an optimal route to satisfy specific constraints and objectives. This problem is applicable to fields such as logistics, travel planning, and vehicle routing. In this paper, we focus on a travel planning scenario with $N$ cities. The traveler must select $K$ cities to visit, starting from a specific city. Each city $c$ provides a profit $p_c$, but the combined profit for visiting multiple cities in one day is a submodular function of these individual profits. The profit of each city changes dynamically over time and resets to 0 once visited. There is a maximum daily travel distance $d_{\max}$, with penalties for exceeding it. The goal is to visit the selected cities within $T$ days and return to the starting point, maximizing overall travel profits.

Specifically, the initial profit of each city follows from a uniform distribution $U(0.5, 1)$ and updates as follows:

$$p_{c,t+1} = p_{c,t} + \xi_{c,t+1}, \quad \text{for } t \geq 0, \tag{15}$$

where $\xi_{c,t+1} \sim U(-0.05, 0.05)$. For a set of cities $C$, the combined profit in one day is defined as

$$f(C) = \sum_{c \in C} p_c - \eta(\max(|C| - 1, 0))^2, \tag{16}$$

where $\eta$ is a coefficient set to 0.1 in our experiments. Exceeding the maximum daily travel distance $d_{\max}$ results in a penalty proportional to the excess distance (we choose 5). The $d_{\max}$ we set is $1.5 \max_{i,j} d_{ij}$, where $d_{ij}$ is the distance of city $i$ and $j$. The datasets used in this problem (coordinates of the city) are also randomly generated using the `random.rand()` function from Python's Numpy library with a scale of 100.

**MDP formulation** For the higher level, the state includes: the coordinates of each city, their profits, the current city, and the starting city. The option is the same as in the AIM problem. The reward is the combined profits of the selected cities at the current time step. The state transition involves the update of profits as described in Equation 15 and the change of the current city.

For the lower level, the state remains consistent with the higher level but also includes the option chosen by the higher level. The action is to select a city. The marginal reward is simply the profit of the selected city, and the immediate reward can be obtained as described in Equation 4. Note that there is no propagation process, and the expected profit of each city does not change. Therefore, we can set $r_{t,\emptyset}^{II}$ to 0. The state transition only involves the change in the current node and is deterministic.

The definitions of MDPs for these two problems are also shown in Table 5, and we will use the framework proposed in this paper to address these problems.

Table 5: MDPs for AIM and RP problems

| Problem | Layer | State | Option/Action | Reward |
|---|---|---|---|---|
| AIM | Higher | active and inactive | choose a budget $k$ | number of new active nodes |
| | Lower | active and inactive, option | select a node | scaled marginal reward of choosing a node |
| RP | Higher | coordinates, profits | choose a budget $k$ | combined profit of $k$ cities |
| | Lower | coordinates, profits, option | select a city | scaled profit |

## C.4 EXPERIMENT HYPER-PARAMETER

To ensure the stability of the learning process in both problems, we employ Double Deep Q-Networks with an update frequency of 10 episodes. The $\epsilon$-greedy strategy we use is initialized with $\epsilon = 0.9$ and decays by a factor of 0.98 after each single training, with a minimum threshold of 0.1. We train each model for 20 epochs and 10 episodes.

All experiments on two COPs are implemented in Python and conducted on two machines. One is NVIDIA GeForce RTX 4090 GPU and Intel 24GB 24-Core i9-13900K. The other is NVIDIA V100 GPU and Inter 256 GB 32-Core Xeon E5-2683 v4.

Table 6: Experiment hyper-parameters for both problems

| Parameter | AIM | RP |
|---|---|---|
| Discount factor ($\gamma$) | 0.99 | 0.99 |
| Learning rate (higher-layer) | $10^{-3}$ | $10^{-2}$ |
| Learning rate (lower-layer) | $10^{-4}$ | $10^{-4}$ |
| Batch size | 32 | 32 |
| DDQN update frequency | 10 episodes | 10 episodes |
| Initial $\epsilon$ in $\epsilon$-greedy | 0.9 | 0.9 |
| Decay factor for $\epsilon$ | 0.98 | 0.98 |
| Minimum $\epsilon$ threshold | 0.1 | 0.1 |
| Epochs | 20 | 20 |
| Episodes | 10 | 10 |
| Hardware | NVIDIA GeForce RTX 4090, Intel 24GB 24-Core i9-13900K, NVIDIA V100, Intel 256GB 32-Core Xeon E5-2683 v4 | |

## C.5 USING T-TEST FOR EVALUATION

To evaluate the performance of our model, we use a t-test rather than standard deviation. This approach is particularly suitable given the presence of both aleatoric and epistemic uncertainties. Typically, we conduct multiple experiments to calculate the mean and standard deviation. Through this process, the epistemic uncertainty—arising from limited data—can be mitigated according to the law of large numbers. However, aleatoric uncertainty, which stems from inherent stochasticity in the environment, is irreducible and contributes to natural variability. Consequently, the standard deviation in such scenarios primarily reflects aleatoric uncertainty. In our stochastic environment, this metric alone provides limited insight into the true performance of the model.

# D ADDITIONAL RESULTS

## D.1 CUMULATIVE REWARD

## D.2 RESOURCE ALLOCATION

Figure 6– 8 shows the budget allocation for AIM problem. The last three days have been ignored due to none resource allocation.

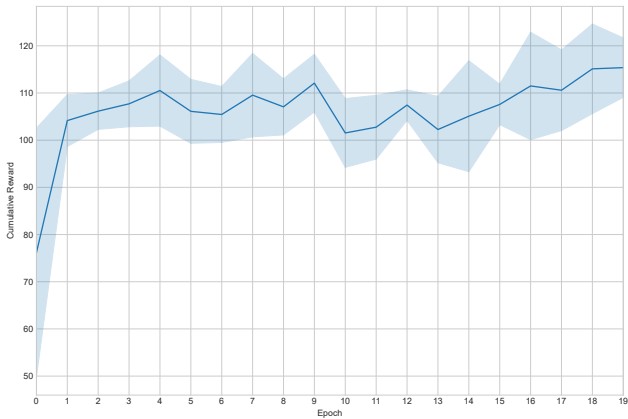

Figure 5: Cumulative reward during training for AIM $T = 10, K = 20$

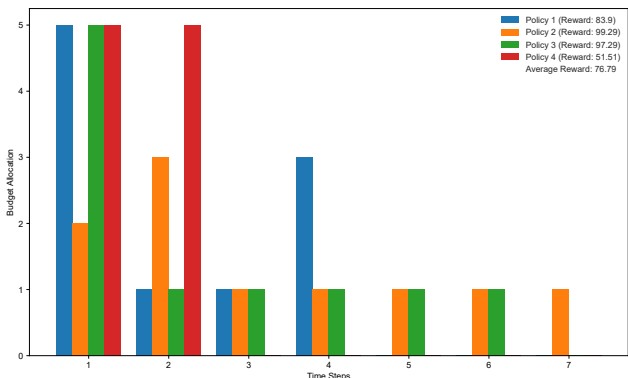

Figure 6: Budget allocation for $T = 10, K = 10$

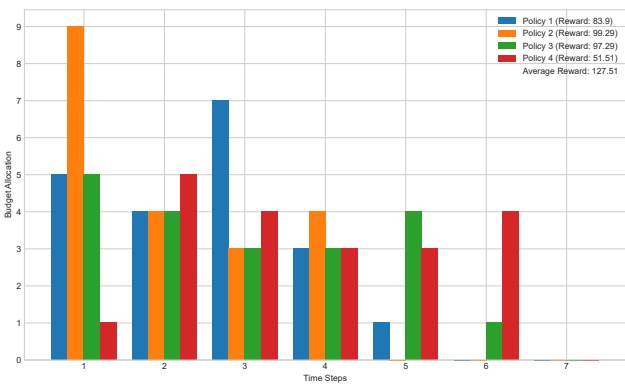

Figure 7: Budget allocation for $T = 10, K = 20$

Table 7: RP: Generalization to larger graphs. All cases have p-values $\leq 0.05$.

|  | 200 | 400 | 700 | 1000 |
|---|---|---|---|---|
| **WS-option** | **14.45** | **13.48** | **14.49** | **14.23** |
| greedy | 12.41 | 10.74 | 11.37 | 12.04 |
| GA | 12.62 | 10.96 | 11.37 | 11.77 |

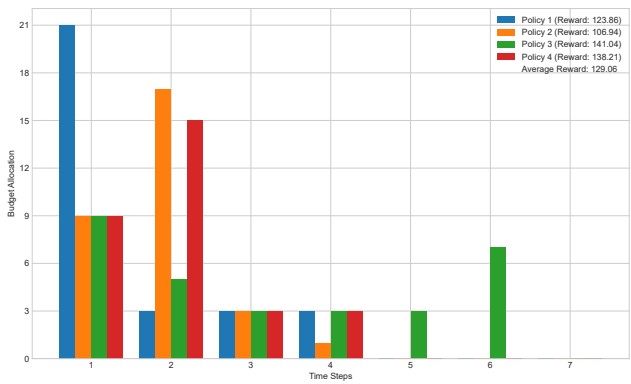

Figure 8: Budget allocation for $T = 10, K = 30$

## D.3 GENERALIZATION TO LARGER GRAPHS

## D.4 EVALUATION ON THE REAL-WORLD DATA

Real-world datasets often exhibit certain patterns, such as clustering characteristics. These patterns are easier for our RL-based algorithm to recognize, which can, in turn, achieve even better results. To demonstrate this, we evaluate our model on *power2500* dataset, which has 2500 nodes.

| Method | T=10, K=20 |
| --- | --- |
| **WS-option** | **496.37** |
| average-degree | 314.49 |
| average-score | 317.16 |
| normal-degree | 449.30 |
| normal-score | 462.54 |
| static-degree | 350.84 |
| static-score | 393.10 |

Table 8: Performance comparison of methods on the *power2500* dataset (T=10, K=20).

The results in Table 8 show that our RL-based algorithm (WS-option) significantly outperforms baseline methods, especially under scenarios characterized by clustering patterns. This demonstrates the model's capability to exploit structural properties in real-world datasets.

## D.5 ABLATION STUDY FOR SLEEP STAGE WITH DIFFERENT TRAINING TIME

| Method | T=10, K=20 |
| --- | --- |
| WS-option (sleep with 1/2 $N$) | 118.56 |
| WS-option (sleep with 1/3 $N$) | 104.84 |
| WS-option (sleep with 2/3 $N$) | 118.59 |
| average-degree | 104.54 |
| average-score | 116.10 |
| normal-degree | 101.50 |
| normal-score | 111.89 |
| static-degree | 105.25 |
| static-score | 118.13 |

Table 9: Performance comparison under different training time for the sleep stage (T=10, K=20).

Note that we use the same random seed as in 4.2. From the results, we observe that ensuring the lower layer converges during the sleep stage is crucial. Specifically, when the model trains with $\frac{1}{2}N$

or $\frac{2}{3}N$, the performance is nearly identical, indicating sufficient convergence. In contrast, training with $\frac{1}{3}N$ leads to significantly degraded performance, demonstrating that insufficient convergence in the lower layer undermines the overall results.

## D.6 ABLATION STUDY FOR SIMPLIFIED OPTION IN THE LOWER LAYER

The experimental results for evaluating the effect of simplifying the option in the lower layer are presented in Table 10. We observe that using a simplified option achieves slightly better performance compared to the non-simplified version (118.56 vs. 116.37). This shows that while the non-simplified option can still yield good results with sufficient iterations, it requires more training for agents to accurately identify termination conditions. Specifically, termination occurs only when the budget equals 0; when the budget is greater than 0, the behavior remains the same across methods.

| Method | T=10, K=20 |
|---|---|
| **WS-option** | **118.56** |
| WS-option (non-simplified option) | 116.37 |
| average-degree | 104.54 |
| average-score | 116.10 |
| normal-degree | 101.50 |
| normal-score | 111.89 |
| static-degree | 105.25 |
| static-score | 118.13 |

Table 10: Ablation study results for simplified options in the lower layer (T=10, K=20).

