# OpenReview forum: "Sequential Stochastic Combinatorial Optimization Using Hierarchal Reinforcement Learning"
_ICLR.cc/2025/Conference — ICLR 2025 Poster_

### Official Review · Reviewer_R7Hz · 2024-10-31

**Soundness:** 2
**Presentation:** 2
**Contribution:** 2
**Rating:** 6
**Confidence:** 4

**Summary:**

This work proposes WS-option, a hierarchical RL approach for sequential stochastic combinatorial optimisation (SSCO). The setting is formulated as a 2-layer MDP, with the higher level policy - trained using an MC approach - learning the budget allocation, while the lower level policy - trained using Q-learning - deciding on the node selection. The key idea, denoted as wake-sleep (WS), is to avoid the adversarial dynamics and instability during training, by fixing the higher level policy for the first half of the training procedure.  The approach is evaluated in two settings, adaptive influence maximisation and route planning.

**Strengths:**

The work provides a formulation of sequential stochastic combinatorial optimisation problems as a 2-layer MDP, facilitating the application of RL methods for this setting. It also proposes a hierarchical RL framework, the WS-option, for solving such settings.

**Weaknesses:**

While the work has some merits in terms of problem formulation under the HRL framework, I argue it lacks a sufficient algorithmic contribution, a comprehensive evaluation and remains generic in many of the implementation details and experiments. I provide below more details and raise also questions that would allow a better comprehension of the work.

The provided details are insufficient to understand and reproduce the work. As far as I understand, Double DQN is used for both policies, but the higher level uses an MC update method, with an action-in network, trained with offline trajectory data gathered by the average budget fixed policy. Providing the complete neural architecture details, number of layers and sizes would improve this issue. Also please see Q3.

I am not sure the convergence analysis for the tabular case is complete and correct, since it does not discuss the dependency between the Q-value of the two layers in terms of how the reward is defined. Eq. 4 claims consistency between layers for the reward definition, but this is not further discussed or supported. See Q2 for further questions on the reward definition. Furthermore,  lines 316-317 state that the Q-functions of the layers are independent, but as detailed in the proof of Theorem 1, this is not the case.

For the experiments, it is not clear in the AIM case, which settings from [1] are used and how the results compare to the ones in the referenced work. I also argue that the baseline selection is weak, only looking at fixed or heuristic based methods. Similar remark stands for the RP case, with no details or references provided for the GA approach. See Q4.

[1] Guangmo Tong, Ruiqi Wang, Zheng Dong, and Xiang Li. Time-constrained adaptive influence maximization. IEEE Transactions on Computational Social Systems, 8(1):33–44, 2020.

Minor remarks:
- I advise to avoid using abbreviations without introduction (e.g., WS, MC, TD lines 88-96, HER line 150)
- typos: adpative (line 219), MD (line 365)

**Questions:**

Q1. The problem formulation proposes to learn a higher level policy for the budget allocation and a lower level policy for the node selection. I failed to understand, given the simplification of the option from the actual budget to a binary value, how the option termination is taking place. Or in other words, how is the connection between the budget and sequence of selected nodes being made in the activation/propagation process, especially since in the lower level the nodes are deterministically activated.

Q2. Secondly, I was also not able to properly grasp the marginal reward calculation (Eq. 4). Could you please provide additional details on the simulation averaging and scaling procedure and its importance on the training stability. Also, how does this definition ensure consistency between the MDPs?

Q3. Could you provide more details and a reference for the action-in architecture and its role in generalisation?

Q4. Could you provide a motivation for the baseline selection and also explain how the results compare to Tong et al. [1].

---

> ### Author Response · Authors · 2024-11-22
> **Response to Review**
>
> We generally appreciate your comments.  However, we would like to emphasize that our primary contribution lies in the development of the HRL framework for SSCO problems—an area that, to the best of our knowledge, has not been explored before. To ensure stability and convergence, we have devoted significant effort to various aspects of the framework design, including (but not limited to) the introduction of null actions, lower-layer reward and option design, layer-wise learning method selection, and the wake-sleep training procedure. Furthermore, we provide a convergence analysis and briefly show its connection to Stackelberg games. Nonetheless, we acknowledge certain limitations and will do our best to address your concerns.
>
> W1: In fact, we provided the details of the neural network we use. See Appendix C.1.
>
> W2: See general response question 5. Lines 316-317 state that they are highly interdependent instead of "independent".
>
> W3: Our choice of $T$ and $K$ in the AIM problem is based on [1]. Additionally, the baseline for the budget allocation policy also comes from their paper. While we evaluate our model on different graphs, we utilize their methods as baselines for comparison. Anyway, as mentioned in our general response, we will provide RL-based baselines in the near future to strengthen our evaluation.
>
> Q1: See general response Q3. The higher-layer decision, i.e., the budget, provides a termination condition for the lower layer. This enables the lower layer to determine when to stop the node selection process. The reason the lower layer is deterministically activated is that this is the basic setting of the AIM problem: once a node is chosen as a seed, it becomes activated. After the node selection process terminates, the agents interact with the environment, receive rewards, and transition to the next state.
>
> Q2: See general respnse 4. We do simulation for $R_t(s^I_t, A_t\cup\{a_{t,v}\})$ and $R_t(s^I_t, A_t)$. It means, after we choose the action set like $A_t$, we can peform a state transition and then get this reward. We simulate this for 10 times to get the average value.
>
> Q3: We use an action-in architecture for the following reasons 1) For different tasks, the total budget varies. Adopting an action-out architecture would mean that the total budget becomes a network parameter, which limits the ability to generalize to tasks with different budgets. 2) After selecting a portion of the budget, subsequent decisions can be made based on the remaining budget. However, with an action-out architecture, the input would include impossible actions, even if a mask is applied, making it less efficient. 3) Finally, the action-in structure is inherently more suitable and effective for our objective of achieving generalization across tasks.
>
> Please refer to slide 7 of Lecture 6: Value Function Approximation, from David Silver’s lecture on Reinforcement Learning.
>
> Q4: First, these two examples, AIM and RP, denote the case with high stochasticity and low stochasticity, respectively. The baselines of AIM comes from [1]. It can be applied to RP. However, since RP is a low stochasticity problem, we use better baselines. For example, GA can usually find the global optimal solution is deterministic case. So we use it in the low-stochasticity case. However, it cannot be applied to high-stochasticity case like AIM. We evaluate our approach on different datasets (their datasets are too large); however, we did provide a comparison between the performance of our method and their methods.
>
> If our response has addressed some of your concerns, we would highly appreciate it if you could re-evaluate our work and consider raising the score.
>
> References:
> [1] Tong, G., Wang, R., Dong, Z., & Li, X. (2020). Time-constrained adaptive influence maximization. IEEE Transactions on Computational Social Systems, 8(1), 33-44.

---

> > ### Comment · Reviewer_R7Hz · 2024-11-26
> >
> > First of all thank you for the detailed answers to the raised questions.
> >
> > As you mention above, your primary contribution lies in the development of the HRL framework for SSCO problems. However, as you can see from the concerns, many of the modelling choices were initially left unclear and unmotivated, and it felt more like ad-hoc decisions based on trial and error and engineering the system to work.
> >
> > Systematically grounding each choice via ablation studies for example, if you take an empirical approach, is needed. You have already greatly augmented the supplementary material in this direction and I believe the ablation experiments you mention in the general response will greatly improve the work.
> >
> > I am still not convinced regarding the option simplification choice. I understand the need to reduce the problem dimensionality, but I am not sure that I agree with the claim:  "Due to the nature of this task, we claim that the conditionally optimal lower-layer policy remains similar across different higher-layer policies" or that "The numerical value of the budget does not provide additional information". If this is indeed the case, then the two policies can be trained somewhat separately and the framework you propose is one method of doing so.
> >
> > I will raise the score to reflect that my questions have been clarified.

---

> > > ### Author Response · Authors · 2024-11-26
> > > **Official Comment by Authors**
> > >
> > > We appreciate your decision and will still try to answer your questions.
> > >
> > > > Due to the nature of this task, we claim that the conditionally optimal lower-layer policy remains similar across different higher-layer policies
> > >
> > > Unlike tasks where agents must learn entirely different actions (e.g., cooking in a kitchen versus delivering food in a dining hall), our task focuses solely on node selection. A similar example is the maze-solving problem. While the lower-layer policy may vary slightly under different higher-layer options, we assume shared underlying patterns—a reasonable assumption that enables joint training of two agents with maintained stability.
> > >
> > > > The numerical value of the budget does not provide additional information.
> > >
> > > We may make some confusions here, as we have already responded to the Reviewer E3Wz. The core of this question is how the option framework [1] works. We did not imply that the actual budget value is irrelevant throughout the process. Instead, we claim that after the joint training process (wake stage), the role of the budget in the lower layer is solely to provide the termination condition. In this context, the lower layer relies on the higher layer's option to determine when to stop node selection—this aligns with the mechanism of the option framework. Notably, the budget information is fully incorporated during the training process. When we jointly train the policies of both agents, the budget information is embedded as the higher-layer policy determines budget allocation. Then, what the higher layer conveys to the lower layer is just termination condition.
> > >
> > > The key is to understand what the option means for the lower layer in the option framework. Does this make sense?
> > >
> > > We provide the ablation study for this:
> > > | Method                            | T=10, K=20 |     |
> > > | :-------------------------------- | ---------: | --- |
> > > | WS-option                         |     118.56 |     |
> > > | WS-option (non-simplified option) |     116.37 |     |
> > > | average-degree                    |     104.54 |     |
> > > | average-score                     |     116.10 |     |
> > > | normal-degree                     |     101.50 |     |
> > > | normal-score                      |     111.89 |     |
> > > | static-degree                     |     105.25 |     |
> > > | static-score                      |     118.13 |     |
> > > If we do not simplify the option in the lower layer, we can still achieve good performance with sufficient iterations. The only difference is that agents will require more training to accurately identify termination conditions. Specifically, termination occurs only when the budget equals 0; for cases where the budget is greater than 0, there is no difference in behavior.
> > >
> > >
> > > ## References:
> > > [1] Richard S Sutton, Doina Precup, and Satinder Singh. Between mdps and semi-mdps: A framework for temporal abstraction in reinforcement learning. Artificial Intelligence, 112(1-2): 181–211, 1999.

---

### Official Review · Reviewer_zNgF · 2024-11-03

**Soundness:** 3
**Presentation:** 3
**Contribution:** 3
**Rating:** 6
**Confidence:** 3

**Summary:**

The paper introduces a novel hierarchical reinforcement learning (HRL) framework, WS-option, which addresses the sequential stochastic combinatorial optimization (SSCO) problem. This framework is innovative as it combines budget allocation and node selection in a two-layer option-based structure, which is a creative approach not commonly seen in the prior works.

**Strengths:**

1.The authors have formulated a generic class of SSCO problems, which is an original contribution to the field. This formulation allows for a more structured approach to solving a broad range of combinatorial optimization problems with stochastic and sequential elements.
2.The paper presents a rigorous methodology, with a clear definition of the Markov decision processes (MDPs) for both layers of the HRL framework. The authors have thoughtfully designed the MDPs to capture the interdependencies between the layers, which is crucial for the model's performance.
3.The paper includes a theoretical analysis of the algorithm's convergence, which is a strong point. The theorems provided offer a solid foundation and justify the effectiveness of the proposed approach.

**Weaknesses:**

1.The paper primarily uses synthetic datasets for experiments, which might not fully capture the complexities of real-world scenarios. Including datasets from diverse real-world applications could strengthen the validity of the results.
2.The paper primarily uses synthetic datasets for experiments, which might not fully capture the complexities of real-world scenarios. Including datasets from diverse real-world applications could strengthen the validity of the results.
3.The paper notes that each new problem may require a new graph embedding technique, which adds complexity to the framework's implementation. Further exploration into developing model-agnostic or more universally applicable graph embedding techniques could enhance the framework's usability across different domains.

**Questions:**

1.Will the WS-option framework's performance vary significantly when applied to real-world datasets compared to synthetic ones? Could the authors share any insights or preliminary results if this framework has been tested on real-world data?
2.How does the WS-option framework scale in terms of computational resources and time when applied to extremely large problem instances, such as graphs with millions of nodes?
3.Are there any plans to explore more universal graph embedding techniques that could be applicable across different SSCO problems without the need for customization?
4.Could the authors provide more details on the computational complexity of the WS-option framework, especially when compared to existing methods in the literature?
5.Are there any plans to conduct ablation studies to isolate the impact of the wake-sleep training procedure and layer-wise learning method selection on the overall performance?
6.Will increasing the number of trials and providing confidence intervals for the experimental results further strengthen the conclusions drawn from the data?

---

> ### Author Response · Authors · 2024-11-22
> **Response to review**
>
> Thank you for the insightful comments. We'll try our best to clarify your concerns.
>
> W1-2: Thank you for pointing it out. We acknowledge this is one of our shortcomings. In fact, we do have some real-world datasets, but the large number of nodes results in significant computational overhead. Moreover, real-world datasets often exhibit certain patterns, such as clustering characteristics. These patterns are easier for our RL-based algorithm to recognize, which can, in turn, achieve even better results. Anyway, we will soon supplement the results on real-world data.
>
> W3: The reason why different tasks require different graph embedding techniques is that, compared to traditional CO tasks, SSCO tasks generally involve more complex graph information. As a result, a mroe sophisticated structure is needed to effectively handle four types of information: (1) node information, (2) edge information, (3) topological structure information, and (4) global information. When tackling this task, we attempted to use a unified graph embedding method, but the results were unsatisfactory. For instance, we tried using GCN, which cannot effectively utilize edge information and fails to accurately represent the graph structure for both types of tasks. Developing a generic graph embedding technique might be worthy of a new paper; however, that is not the focus of this work. Nonetheless, our next work involves proposing a graph embedding method tailored for RL Q-value representation based on a graph foundation model. Building on previous works, such as [1], we provide corresponding theoretical proofs to support this approach.
>
> Q1: See W1-2.
>
> Q2: Our work is parallel to the effort in [2], and their techniques can certainly be applied to extend our framework to handle extremely large graphs, such as those with billions of nodes. However, scaling to this level requires significant effort and computational resources, so we are unable to provide further experiments for such large-scale instances at this time.
>
> Q3: See W3.
>
> Q4: Computational complexity in RL literature is indeed challenging to analyze. To the best of our knowledge, there are no existing works that provide a comprehensive complexity analysis for similar frameworks. While we acknowledge the importance of such an analysis, our current focus has been on the practical effectiveness and scalability of the WS-option framework.
>
> Q5: We have already provided ablation stuides for the wake-sleep training procedure and the layer-wise learning method selection. See lines 378-391.
>
> Q6: See Appendix C.5 for the reason why we use t-test instead of mean-standard deviation.
>
> If our response does not effectively address your concerns, please feel free to ask further questions.
>
> References:
> [1] Xu, K., Hu, W., Leskovec, J., & Jegelka, S. (2019). How powerful are graph neural networks?. International Conference on Learning Representations.
> [2] Manchanda, S., Mittal, A., Dhawan, A., Medya, S., Ranu, S., & Singh, A. (2019). Learning heuristics over large graphs via deep reinforcement learning. arXiv preprint arXiv:1903.03332.

---

> ### Author Response · Authors · 2024-11-26
> **Offical Comment by Authors**
>
> Experiments on power2500 dataset
>
> | Method                       | T=10, K=20 |     |
> | :--------------------------- | ---------: | --- |
> | WS-option                        |     496.37 |     |
> | average-degree               |     314.49 |     |
> | average-score                |     317.16 |     |
> | normal-degree                |     449.30 |     |
> | normal-score                 |     462.54 |     |
> | static-degree                |     350.84 |     |
> | static-score                 |     393.10 |     |
>
> The result is consistent with what we said before:
> > Moreover, real-world datasets often exhibit certain patterns, such as clustering characteristics. These patterns are easier for our RL-based algorithm to recognize, which can, in turn, achieve even better results. Anyway, we will soon supplement the results on real-world data.

---

### Official Review · Reviewer_E3Wz · 2024-11-04

**Soundness:** 1
**Presentation:** 3
**Contribution:** 2
**Rating:** 6
**Confidence:** 3

**Summary:**

This paper proposes WS-option, a hierarchical RL (HRL) algorithm to tackle sequential stochastic combinatorial optimization (SSCO) problems. These type of problems require to allocate a fixed budget over multiple timesteps, and solve the problem over these timesteps under allocation constraint. WS-option defines a high-level policy that allocates the budget, and a low-level policy that solves the underlying problem (selecting a combination of nodes of a graph in this case). The algorithm is experimentally validated on 2 problem instances.

**Strengths:**

There are many ways to combine RL with combinatorial optimization, but it seems particularly well-suited in this case, as RL naturally handles the stochastic and sequential nature of SSCO. I believe the RL-based approach envisioned by the authors has high potential. It also seems to adapt well to different problem-size instances, which is not necessarily the case for RL-based methods for combinatorial optimization.

**Weaknesses:**

Nevertheless, I believe that it is insufficient in its current form for the following main reasons:

1. Design choices lack proper justification

     a. The use of HRL here introduces 2 different agents, which introduces nonstationarity due to multiple policies interacting with each other. However, it seems here that a single policy with 2 different action-spaces -- one for the budget allocation, another one for the node selection -- (or one action-space with an action-mask to select only valid actions) could mitigate this issue. The importance of HRL is unclear.

    b. The lower-level MDP includes an option that can either be 0 or 1, which defines if the lower-level policy can continue to select nodes. From my understanding, the state-space does not include the allocated budget defined by the higher-level policy.  But this information should be crucial for the lower-level policy to choose which nodes to select. Now, at each timestep, the lower-level policy does not know if it will be able to continue selecting a node at the next timestep or not. Why resort to an "intermediate" option with 0 or 1, instead of allocating the $o_t^I$ in the state?

    c. The first half of training does not use the higher level policy to stabilize the training of the low-level policy. Are there any preliminary experiments that justify the choice of 1/2 of the training time? How dependent is this on the problem?

    d. The Route Planning problem (Appendix E) updates the profits by sampling U(-0.05, 0.05). What is the reason for this choice? Also, sampling from U(-0.05, 0.05) results in 0 in expectation. Wouldn't this affect the marginal reward? What are the consequences on the sequentiality of the problem (i.e., how much does this change compared to solving everything in one shot)?

    e. The results do not show the standard-deviation, resorting instead to two-sample t-tests. Why? There is no clear justification as to why the standard-deviation is left out.

2. Experimental baselines are non-adaptive

    a. I understand that, since the authors are the first to propose an adaptive budget allocation for SSCO problems, they compare it with some fixed allocation strategies (average, static, normal). But there are no RL baselines, making it hard to assess the added value of the proposed algorithm. A standard RL baseline would also help understanding the need for HRL (e.g., using 2 action spaces as mentioned in 1b).

    b. Even though the baselines are non-adaptive, they are different for AIM and for RP. I fail to understand why, and why the average/normal/static high-level policies could not be used for RP.

**Questions:**

Additional questions:

 1. In table 1, the performance of WS-option seems to be close to the "score" baseline  (for static and average). Are there any insights for this?
 2. It is unclear to me how the simulations are performed. Is it executing the greedy policy 10 times from the current state?
 3. Why are the WS-option scores in table 3 not the same as in table 1? Shouldn't they be the same, since they use the same high-level/low-level policy combination?

---

> ### Author Response · Authors · 2024-11-21
> **Response to review**
>
> We appreciate the insightful comments and questions from the reviewer. Now we try to address concerns from the reviewer.
>
> W1.a: See general response. The detail is provided in Appendix A.1. The biggest problem of single-agent RL is that it cannot deal with the interdependency between two actions, while HRL does.
>
> W1.b: The state space indeed doesn't include the option, since option can be seen as another state. Note that the q-function for the lower-layer is $q^{II}(s, o, a)$, so the option can also be seen as a state. As a result, the "real state" here is $(s,o)$. See general respond for the reason of simplifying the option in the lower layer. The detail is provided in Appendix A.2.
>
> W1.c: Since the higher-layer policy will change in the begining of the training if we don't use a sleep stage, it leads to the unstability of the second layer. See general response and Appendix A.2 for more details of our wake-sleep training procedure. In fact, the 1/2 of training doesn't matter in this case. The key idea is to make the second layer converges first (the reason is provided in Appendix A.2 and A.5), so we can just increase $N$ to achieve it. However, we will also provide an ablation study for different choices like N/3, and show what happens if the second layer doesn't coverge. This is independent of problems since our goal is to make the lower layer converge.
>
> W1.d: This is the basic setting of RP problem, where the profit of each node changes every day. So we add this U just to model the fluctuation of the profit. We can also use like a guassian noise. The choice of the noise doesn't matter. As a result, we choose a simple one because we want a case of low stochasticity compared with AIM.  This is a good point that the expectation is 0. This is also the magic of our null action, which defines the the expected reward when moving to the next day if we do not select any node in the current state.  So if the reward of null action is exactly the expectation of noise here. Hence, it will affect the marginal reward according to our definition. We can consider a case where all the noise is accidentally positive, which means the profit will increase over days. Then we cannot find an optimal solution. Further, the combined profit will decrease as the number of budget increase at time step t. This also leads to a suboptimal solution. See lines 67-73 for our motivation of SSCO problems.
>
> W1.e: See Appendix C.5.
>
> W2.a. We will provide RL baselines soon, although it's not quite suitable for this case.
>
> W2.b. First, these two examples, AIM and RP, denote the case with high stochasticity and low stochasticity, respectively. The methods used in AIM can be applied to RP. However, since RP is a low stochasticity problem, we use better baselines. For example, GA can usually find the global optimal solution is deterministic case. So we use it in the low-stochasticity case. However, it cannot be applied to high-stochasticity case like AIM.
>
> Q1: We ackonwledge that finding a quite strong baseline is difficult. As a result, we design a "score" policy, which is essential a greedy policy. However, we leverage the feature of AIM problem -- the difinition of propogation probability. Based on this, we use the outdegree of one node combined with the indegree of all its inactive neighbors to get a score which is truly the expected influence of that node in two steps. Since the propagation probability is quite small, the influence of one node can just spread for 2 time steps in some cases (for some nodes). So this problem-specific design is indeed a strong baseline.
>
> Q2: Sorry for the confusion. Note that we do simulation for $R_t(s^I_t, A_t\cup\{a_{t,v}\})$ and $R_t(s^I_t, A_t)$. It means, after we choose the action set like $A_t$, we can peform a state transition and then get this reward. We simulate this for 10 times to get the average value. The action samples are obtained when we run the episodes (see Algorithm 2), and then we use the action sample to generate corresponding transitions for the lower layer (see Algorithm 3). When we do the simulation, the choice of actions is already given.
>
> Q3: The reason is we didn't evaluate the model on the same graph. And the score depends on the graph that we test.
>
> Feel free to ask any questions if we haven't make them clear enough.

---

> > ### Comment · Reviewer_E3Wz · 2024-11-25
> >
> > > Note that the q-function for the lower-layer is $q^{II}(s,o,a)$, so the option can also be seen as the state.
> >
> > Algo 2, line 11 clearly states $q^{II} (s_{t,k}^{II}, o_{t,k}^{II}, a) $ ($ o_{t,k}^{II} $ has been added in the revision). I believe my point remains: the budget allocation ($o_{t}^{I}$) is not provided as information to the low-level policy. Only the low-level option ($o_{t,k}^{II}$, a binary value) is included. With only this information, it is impossible for the low-level policy to plan ahead, as it does not know if it has a budget of let's say 5 or a budget of 2 to allocate for the given timestep. This might determine which nodes to select: knowing you only have a budget of 2, you might select other nodes than if you knew you could select 5. Am I missing something? From my understanding, the discussion provided in A.3.1 does not convince me that this information is not useful.
> >
> > > In fact, the 1/2 of training doesn't matter in this case
> >
> > Thank you for this information.
> >
> > > This is a good point that the expectation is 0. This is also the magic of our null action
> >
> > Thank you for clarifying the advantage of the null action wrt the expectation of 0. How restrictive is this? You mention that:
> >
> > > we can consider a case where all the noise is accidentally positive, which means the profit will increase over days. Then we cannot find an optimal solution.
> >
> > Why wouldn't you find an optimal solution? The RL agent perceives a reward at each timestep (which, granted, increases over time), and aims to maximise the sum of rewards. How does this make it impossible to find the optimal solution?
> >
> > > The reason is we didn't evaluate the model on the same graph
> >
> > Does this mean all the results shown in the paper are performed on a single graph for each experiment? How many seeds did you run for each algorithm? Given this, I might have misinterpreted $n=200, n=100$ in the captions of Table 1, 2 respectively. What is $n$?

---

> ### Author Response · Authors · 2024-11-26
> **Official Comment by Authors**
>
> Thanks a lot for your comments. We now understand what's your concerns.
>
> > Am I missing something? From my understanding, the discussion provided in A.3.1 does not convince me that this information is not useful.
>
> One point we didn't make it clear is that when we train the policies of both layers together, we have included the budget information (e.g., the specific value of budget) for both agents. Then, what the higher layer conveys to the lower layer is exactly the termination condition. The is also how the option framework works [1]. Then, we discuss the budget irrelevance in Appendix, which means in the option framework, the higher layer just conveys the termination condition to the lower layer, and the lower layer is solely aware of this termination condition and nothing more.
>
> This question essentially is how the option framework proposed by Sutton [1] works. Although it is not the focus of our work (we just used it), we are happy to address your concern regarding this matter.
>
> We also have the ablation studies for this:
>
> | Method                            | T=10, K=20 |     |
> | :-------------------------------- | ---------: | --- |
> | WS-option                         |     118.56 |     |
> | WS-option (non-simplified option) |     116.37 |     |
> | average-degree                    |     104.54 |     |
> | average-score                     |     116.10 |     |
> | normal-degree                     |     101.50 |     |
> | normal-score                      |     111.89 |     |
> | static-degree                     |     105.25 |     |
> | static-score                      |     118.13 |     |
> If we do not simplify the option in the lower layer, we can still achieve good performance with sufficient iterations. The only difference is that agents will require more training to accurately identify termination conditions. Specifically, termination occurs only when the budget equals 0; for cases where the budget is greater than 0, there is no difference in behavior.
>
> We mentioned
> > we can consider a case where all the noise is accidentally positive, which means the profit will increase over days. Then we cannot find an optimal solution.
>
> to address your concern:
>
> > What are the consequences on the sequentiality of the problem (i.e., how much does this change compared to solving everything in one shot)?
>
> In other words, in this case, if we allocate all the budget at the first time step, we cannot get the optimal solution, as profits will increase over subsequent days.
>
> > Does this mean all the results shown in the paper are performed on a single graph for each experiment? How many seeds did you run for each algorithm?
>
> Although we train the model on different graphs following the same distribution, we test the model on a single random graph (fixed seed, or given real-world data) for each experiment. However, we do not use the same graph for all experiments. Does this make sense? The $n$ represents the graph size.
>
> ## References:
>
> [1]  Richard S Sutton, Doina Precup, and Satinder Singh. Between mdps and semi-mdps: A framework for temporal abstraction in reinforcement learning. Artificial Intelligence, 112(1-2): 181–211, 1999.

---

> > ### Author Response · Authors · 2024-11-26
> > **Official Comment By Authors: Ablation studies for the training time of the sleep stage**
> >
> > | Method                       | T=10, K=20 |     |
> > | :--------------------------- | ---------: | --- |
> > | WS-option (sleep with 1/2 N) |     118.56 |     |
> > | WS-option (sleep with 1/3 N) |     104.84 |     |
> > | WS-option (sleep with 2/3 N) |     118.59 |     |
> > | average-degree               |     104.54 |     |
> > | average-score                |     116.10 |     |
> > | normal-degree                |     101.50 |     |
> > | normal-score                 |     111.89 |     |
> > | static-degree                |     105.25 |     |
> > | static-score                 |     118.13 |     |
> >
> > We can see that if we guarantee the lower layer converges in the sleep stage, we get almost the same gain (sleep with 1/2 and 2/3 N). In contrast, if the lower layer doesn't converge, the performance is bad (sleep with 1/3 N).
> >
> > As we said before, the choice of 1/2 doesn't matter --- we just need to make sure the lower layer converges in the sleep stage. Therefore, we can just choose larger $N$.

---

> > > ### Comment · Reviewer_E3Wz · 2024-11-29
> > >
> > > I am aware of the option framework. Maybe if I frame it differently, you will better understand my point.
> > > An option $o$ is a tuple consisting of 3 components $(I, \pi, \beta)$:
> > >
> > > - $I \subseteq S$ is the initiation set. $o$ is only available in state $s$ if $s \in I$.
> > > - $\pi : S \times A \rightarrow [0,1]$ is the policy executed under option $o$.
> > > - $\beta : S \rightarrow [0,1]$ is the termination condition.
> > >
> > > Note that each of the 3 components, $I, \pi, \beta$ operate under *the same state-space*. For each state $s$, we can execute the termination condition $\beta$: $s$ thus contains all the necessary information for the option to decide if it is time to stop. In the current problem formulation,
> > > $$\beta(s) =
> > > \begin{cases}
> > > 1 & (K_t = 0) \\\\
> > > 0 & (K_t > 0)
> > > \end{cases}$$
> > >
> > > $s = \\{X, g\\}$ does not contain $K_t$, and so $\beta(s)$ cannot properly decide if the option should terminate or not. To me, *some information is missing in the state-space* (in this case $K_t$). I would love to be proven wrong, but stating that "this is how the option framework works" is somehow not convincing (I hope I made clear why).
> > >
> > > ---
> > >
> > > Concerning
> > > > we test the model on a single random graph (fixed seed, or given real-world data) for each experiment. However, we do not use the same graph for all experiments.
> > >
> > > Thank you for this clarification. I am actually quite surprised by this. The works that are cited in the introduction [24, 39, 14, 5, 26, 6] all evaluate on a held-out test set of multiple graphs (typically 1000) to ensure the results are representative. Is there a reason for evaluating on a single graph?

---

> > > > ### Author Response · Authors · 2024-12-01
> > > > **Official Comment By Authors**
> > > >
> > > > Thank you for your patience in discussing the details with us. We are happy to further elaborate on our framework.
> > > >
> > > > We are not just saying, "this is how the option framework works." More importantly, we aim to explain how the option framework is applied in our case. There are several points to clarify:
> > > >
> > > > - We have explained that the lower layer takes the state-option as input, where $\beta(s)$ is provided by the higher layer and determines when to stop (0 or 1). This directly addresses your concern that "$s = \set{X, g\}$ does not contain $K_t$, and so $\beta(s)$ cannot properly decide if the option should terminate." The decision of "when to stop" is entirely made by the higher layer, which also generates $K_t$ as part of its output. Therefore, the higher layer has full knowledge of $K_t$.
> > > >
> > > > - $K_t$ is **INDEED** included in our framework. It seems there’s a misunderstanding that $K_t$ must directly serve as an input for the lower layer when computing the Q-value, as suggested by "some information ($K_t$) is missing in the state-space." However, we emphasize that $K_t$ primarily functions to **signal when to stop**. This aligns with the termination mechanism in the option framework, where **the higher layer learns the termination condition**.
> > > >
> > > >   In our case, we simplify $K_t$ to a binary indicator and pass it to the lower layer. This approach is valid because $K_t$ specifically serves as the termination condition, a role traditionally handled by the higher layer in the option framework. **Our results (see previous comments or Appendix D.6) strongly support this design**.
> > > >
> > > > - A key misunderstanding seems to be the belief that we need to determine $K_t$ first and then use it as an input to the lower layer. This is not true. The process works in the opposite direction: the lower layer takes "whether to stop" as an input, and the higher layer is trained to achieve this behavior. Whether the higher layer explicitly outputs $K_t$ is not critical; the essential point is designing how the termination function is conveyed to the lower layer.
> > > >
> > > > - Why does this design work? A critical factor is the **joint training of both layers during the wake stage**. This ensures that the policies of both layers are interdependent. For instance, the higher layer’s output ($K_t$) inherently considers the behavior of the lower layer. Consequently, the lower layer’s decision-making does not lose any information, as the joint training integrates the actions of both layers. Viewing the two agents in isolation would ignore this interdependence and lead to a misunderstanding of how the system functions.
> > > >
> > > > It's not surprising that evaluation in stochastic combinatorial problems, such as disease propagation [41] and AIM [48], is often conducted on specific datasets (graphs). Unlike deterministic problems [24, 26], stochastic combinatorial problems have inherent randomness. Different graphs exhibit varying levels of stochasticity, which significantly affects results. Evaluating on multiple graphs and computing an average score can be misleading due to these differences in stochasticity.
> > > > Moreover, ensuring a significant p-value in a t-test requires running a large number of experiments (e.g., 5k–15k) on a single graph, which often takes several days. Scaling this to evaluate over 1,000 graphs, as done in deterministic cases [24], is impractical. Therefore, our approach aligns with standard practices for evaluating stochastic combinatorial problems.

---

> > > > > ### Comment · Reviewer_E3Wz · 2024-12-02
> > > > >
> > > > > Thank you for taking the time to elaborate on your framework. I believe this has (not without efforts) clarified the role of $\beta(s)$. I will raise my score, considering that this, combined with the study over the sleep stage parameter and the non-simplified option study have helped justify the use of HRL.

---

### Author Response · Authors · 2024-11-21
**General response**

Dear reviewers,

Thank you for your valuable feedback and recognizing the following strengths of our paper:

* A generic formulation of sequential stochastic combinatorial optimization problems (Reviewers R7Hz and zNgF).
* The RL-based or HRL framework, WS-option, to solve SSCO problems (all the reviewers).

We tried to addressed and incorporated the feedback to strengthen our work. Below are the main changes of the updated version:

* Theoretical insights of our design (Appendix A). Specifically, we answers 5 questions: 1) Why did we use HRL? 2) Why does wake-sleep option framework work? 3) Why did we simplify options in the lower layer to a binary indicator? 4) How to design lower-layer reward? 5) What does our convergence analysis matter?
* Reason for using t-test (Appendix C.5).
* Clearer description of Theorem 2 (line 401).
* A fix to notations in Algorithm 2 (lines 306 and 308).

Specifically, we also answer some questions here:

1. Why did we use HRL instead of traditional RL with 2 different action-spaces? (See Appendix A.1 for details)

   1. This SSCO problem is essentially a bi-level optimization problem. Given its hierarchical dependency, it is natural and intuitive to employ a hierarchical learning algorithm such as HRL to address such problems effectively.
   2. Single-agent RL cannot capture this interdependency between layers, as well as the interdependent objective.
2. Why did we propose this wake-sleep option framework? (See Appendix A.2 for details)

   1. The core idea for solving this bi-level optimization (SSCO) problem is to decompose it into two separate single-level optimization problems.
   2. The sleep stage aims to find the conditionally optimal solution for the lower layer given the higher layer policy.
   3. The wake stage aims to optimize both layers jointly under the observation that the conditionally optimal lower-layer policy for any higher-layer policy is similar, since the task of lower layer is just to select nodes. Then this joint training further refines both the higher-layer policy and the lower-layer policy, ensuring that the global optimal solution is achieved.
3. Why did we simplify the option in the lower layer? (See Appendix A.3 for details)

   1. What the option means for the lower layer is essentially the termination condition, as defined in the option framework. The numerical value of the budget $K_t$ does not provide additional useful information beyond determining the number of node selections. Therefore, conveying the exact budget value to the lower layer is unnecessary and does not contribute to the termination decision. By representing options as binary indicators, we clarify whether the lower layer should continue or terminate, enhancing the interpretability of the policy.
   2. Reducing the option to a binary indicator allows the lower layer to focus solely on a straightforward termination condition, without needing to interpret specific budget values. This simplification minimizes the complexity of the learning task, leading to faster convergence and more stable learning processes.
4. How did we design the lower-layer reward? (See Appendix A.4 for details)

   1. The key is the total reward of the lower layer for each time step $t$ is the same as the reward of the higher layer, which means they share the same objective and reward function $R(s^I_t, o^I_t, a_t^{II})$.
   2. When the lower-layer actions are decomposed into sequential sub-actions (e.g., selecting one node at a time), the individual rewards for each sub-action must sum up to the total reward $R(s^I_t, o^I_t, a_t^{II})$.
5. Why does our convergence analysis matter?(See Appendix A.5 and A.2 for details)

   1. We try to decouple layer dependencies during training by alternating between the two optimization processes. This iterative approach leverages the individual convergence of each layer, as proven in Theorems 1 and 2, to ensure the overall framework converges.
   2. Based on this idea, we further simplify the training process to a wake-sleep training procedure. In this case, the setting indeed is the same with a Stackelberg game setting [1], where the follower can always provide the best response.
   3. We also claimed that this convergence analysis is not a global convergence analysis. Instead, it's the convergence of each optimization process. By using this iterative training procedure, it provides a convergence of both layers implicitly.

Furthermore, we are still running some extra experiments. We will provide all the results soon (approximately within 2 days):

* RL baselines
* Ablation studies for 1/2 training time.
* Test on real-world data
* Ablation studies for lower-level options (budget to binary indicator)

Kind regards,

Authors

References:

[1] Zhao, G., Zhu, B., Jiao, J., & Jordan, M. (2023, July). Online learning in stackelberg games with an omniscient follower. In *International Conference on Machine Learning* (pp. 42304-42316). PMLR.

---

> ### Author Response · Authors · 2024-11-28
> **Official Comment by Authors**
>
> We have updated the paper with following experiments:
> - Ablation studies for 1/2 training time.
> - Test on real-world data
> - Ablation studies for lower-level options (budget to binary indicator)
>
> We also add the RL-baseline here.
>
> | Method          | T=10, K=20 |     |
> | :-------------- | ---------: | --- |
> | **WS-option**   | **118.56** |     |
> | single-agent RL |     108.24 |     |
> | average-degree  |     104.54 |     |
> | average-score   |     116.10 |     |
> | normal-degree   |     101.50 |     |
> | normal-score    |     111.89 |     |
> | static-degree   |     105.25 |     |
> | static-score    |     118.13 |     |
>
> This result serves as a baseline using single-agent RL. We do not employ a single agent with two distinct action spaces because these action spaces are interdependent. Instead, our single-agent RL framework is built upon our lower-layer agent, allowing the agent to select multiple actions (nodes) at each time step. We incorporate a null action as the termination node, indicating the end of the current time step.
>
> We use the same number of training epochs for both approaches; however, it is evident that the performance of single-agent RL is worse than that of our HRL framework. This discrepancy highlights an inherent advantage of HRL: enhanced sample efficiency. HRL achieves this by decomposing the action space into smaller, more manageable subsets, thereby facilitating more efficient training. To improve the performance of single-agent RL, it may be necessary to utilize a more sophisticated network architecture. The SSCO problems are inherently complex. While single-agent RL can be applied, it requires extra effort to carefully design and tailor its architecture. Moreover, single-agent RL is often not generalizable across different complex tasks. In contrast, the HRL framework provides a more interpretable structure and demonstrates superior adaptability to complex tasks. As discussed in Appendix A.1, these types of problems are essentially bi-level optimization problems, making HRL a natural choice for addressing them.
>
> Kind regards,
>
> Authors

---

### Meta-Review · Area_Chair_j1NP · 2024-12-09

**Metareview:**

This paper considers using reinforcement learning (RL) techniques to solve the sequential stochastic combinatorial optimization (SSCO) problems, where one needs to first decide the budget allocation for each step and then select a set of nodes for each time step. As a sequential decision-making model under uncertainty, the authors naturally introduce an RL framework to the SSCO problems. A novel wake-sleep training procedure and a hierarchical RL framework is proposed in this paper to solve the problem. Experimental validation has been done on the propagation problem and the route planning problem. In particular, during the rebuttal period, the experimental validation has been significantly enhanced by adding RL baselines, ablation studies for 1/2 training time, real-world data experiments, and ablation studies for lower-level options, making the empirical studies more convincing. Due to these contributions, we decide to accept this paper.

**Additional Comments On Reviewer Discussion:**

Besides the minor issue that are relatively easy to justify, the main concerns from the reviewers are 2-fold.

1. The motivation and theoretical insights of the proposed method. E.g. Why using HRL? Why using wake-sleep option? Why simplifying lower layer options? And so on.

The authors have provided complete answers to these questions and have added a considerable amount of content in the Appendix to justify these points.

2. Questions on the experimental setting. E.g. Why not using real-world data? Why select 1/2 training time? Why not compare with RL-based method? And so on.

The authors have provided a full list of additional experimental result to justify these points.

Overall, the authors have nicely addressed the concerns from the reviewers during the rebuttal.

---

### Decision · Program_Chairs · 2025-01-22

Accept (Poster)